# SHAPE OF THOUGHT: WHEN DISTRIBUTION CAN MATTER MORE THAN CORRECTNESS IN REASONING TASKS

## ABSTRACT

We present the surprising finding that a language model's reasoning capabilities can be improved by training on synthetic datasets of chain-of-thought (CoT) traces from more capable models, even when all of those traces lead to an incorrect final answer. Our experiments show this approach can yield better performance on reasoning tasks than training on human-annotated datasets. We hypothesize that two key factors explain this phenomenon: first, the distribution of synthetic data is inherently closer to the language model's own distribution, making it more amenable to learning. Second, these 'incorrect' traces are often only partially flawed and contain valid reasoning steps from which the model can learn. To further test the first hypothesis, we use a language model to paraphrase human-annotated traces – shifting their distribution closer to the model's own distribution – and show that this improves performance. For the second hypothesis, we introduce increasingly flawed CoT traces and study to what extent models are tolerant to these flaws. We demonstrate our findings across the MATH, GSM8K, and Countdown datasets using various language models ranging from 1.5B to 9B across Qwen, Llama, and Gemma models. Our study shows that curating datasets that are closer to the model's distribution is a critical aspect to consider. We also show that a correct final answer is not always a reliable indicator of a faithful reasoning process.

## 1 INTRODUCTION

Large language models (LLMs) have made rapid progress on reasoning tasks, often using supervised fine-tuning on chain-of-thought (CoT) traces (Guo et al., 2025a; Bercovich et al., 2025; Ye et al., 2025). A common assumption in this line of research is that correctness is the primary determinant of data quality: the more correct a dataset is, the better it should be for training (Ye et al., 2025; Muennighoff et al., 2025). This assumption has guided the construction of widely used reasoning datasets, which typically rely on heavy human annotation (Hendrycks et al., 2021a; Cobbe et al., 2021) or filtering of model outputs with rule-based verifiers, that validate the CoT traces via final-answer checking (Guo et al., 2025b; Zelikman et al., 2022a). In this work we consider three broad categories of reasoning datasets that could be used to finetune a model: (1) Human-annotated and written traces — fully correct and carefully verified. These are treated as the gold standard. However, these traces may be far from the model's distribution. (2) Synthetic traces generated from more capable models, typically from the same family, leading to correct answers — often filtered using rule-based verifiers that check only the final solution. These traces are closer to the distribution of the model being finetuned, but their reasoning steps may still be partially flawed. (3) Synthetic traces generated from more capable models, typically from the same family, leading to incorrect answers — generally discarded in existing pipelines. Yet these traces are inherently close to the distribution of the model being finetuned, and many contain correct intermediate steps, valid decompositions, or useful patterns, despite reaching the wrong final answer.

Categories (1) and (2) are consistently favored in prior work Ye et al. (2025); Guo et al. (2025b); Zelikman et al. (2022a). Human-annotated data is trusted for correctness, while model-generated correct-answer traces provide scale. Category (3), however, is largely ignored under the assumption that incorrect answers imply poor reasoning. Works such as (Setlur et al., 2024; Aygün et al., 2021) propose to utilize incorrect traces for training better models in a contrastive setting or to train better verifiers. However, whether these traces can directly be useful for improving math reasoning has not been thoroughly tested.

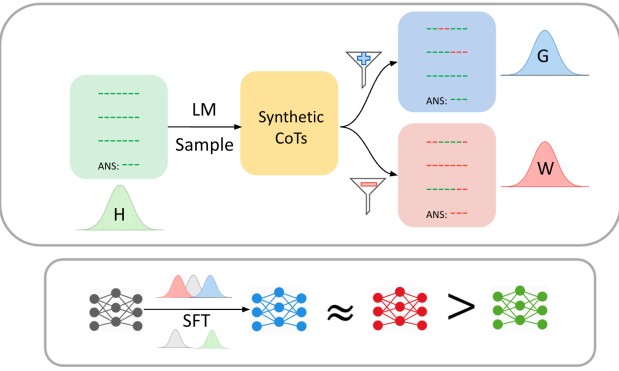

**Figure 1: Fine-tuning on synthetic CoTs, even those with incorrect final answers, can outperform training on human-written data**. We generate two synthetic datasets using a stronger model: **G**, containing CoT traces with correct final answers, and **W**, containing traces with incorrect final answers. This plot shows that fine-tuning a weaker model on both G and W datasets leads to higher downstream accuracy compared to the baseline of training on human-written CoTs (**H**)

We therefore pose two questions: (1) *Can model-generated CoT traces that lead to incorrect final answers still directly help models learn to reason better — and if so, why?* (2) *Should we prioritize fully correct human-written traces that may lie further from the model's output distribution, or model-generated traces that are closer to this distribution - even if they are imperfect?*

To investigate the above questions, we conduct a systematic study of supervised fine-tuning (SFT) on reasoning traces across categories (1), (2), and (3) above. Our experiments cover multiple reasoning benchmarks — MATH (Hendrycks et al., 2021b), GSM8K (Cobbe et al., 2021), and Countdown (Pan et al., 2025) — and three model families ranging from 1.5B to 9B parameters (Gemma (Team et al., 2024), Llama (Grattafiori et al., 2024), Qwen (Qwen et al., 2025)). We generate reasoning traces for Category (2) and Category (3) using the same or more capable language models (LMs). Surprisingly, we find that training with Category (3) data can improve reasoning performance, even more than using human-written correct traces. We also show that paraphrasing human solutions with an LLM can also improve performance by bringing them closer to the model's distribution. Finally, we design experiments to introduce completely flawed CoTs in our datasets successively to reveal how tolerant models are to errors before the performance starts to diminish.

While recent approaches prioritize correctness, our results highlight an underexplored dimension: closeness to the model's distribution can matter as much as, or even more than, correctness.

We summarize the main contributions of our work as follows:

- We show that model-generated CoT traces leading to incorrect final answers (Category 3), which are typically discarded, can improve reasoning performance when used for supervised fine-tuning.
- We demonstrate that training data closer to the model's output distribution, even if imperfect, can be more effective than correct human-written traces that might be further from model's distribution.
- We progressively degrade CoTs to quantify how much incorrect reasoning a model tolerates, before performance degrades — providing insights into the robustness of learning from imperfect data.
- We paraphrase human data to better match the model distribution, improving reasoning scores.
- Qualitatively, we analyse CoT traces generated from the models and show that final answer checking is not the most reliable or holistic way to evaluate CoTs.

## 2 BACKGROUND AND PRELIMINARIES

In this section, we provide the background on improving reasoning in LLMs. We begin by formalizing LLMs and Supervised Fine-Tuning (SFT) as the learning paradigm. We then show how large neural networks can learn from noisy data and, finally, draw a parallel to the regularization phenomena observed in human cognitive development, which serves as the inspiration for our analysis.

## 2.1 LARGE LANGUAGE MODELS AND SUPERVISED FINE-TUNING

An autoregressive Large Language Model, parameterized by $\theta$, models the probability of a text sequence $\mathbf{y} = (y_1, y_2, \ldots, y_T)$ conditioned on an input context $\mathbf{x}$. This is achieved by factorizing the joint probability into a product of conditional probabilities for each token:

$$p_\theta(\mathbf{y} \mid \mathbf{x}) = \prod_{t=1}^{T} p_\theta(y_t \mid \mathbf{x}, \mathbf{y}_{<t}) \tag{1}$$

where $\mathbf{y}_{<t} = (y_1, \ldots, y_{t-1})$ represents the preceding tokens. The probability of the next token $y_t$ is typically derived from a softmax function applied to the model's output logits, often modulated by a temperature parameter $\gamma$ to control the randomness of the generation.

While pre-trained LLMs possess broad capabilities, adapting them to specific tasks is commonly achieved through **Supervised Fine-Tuning (SFT)**. SFT is a process that updates a model's parameters using a labeled dataset $\mathcal{D} = \{(\mathbf{x}, \mathbf{y})\}$ of input-output pairs. The training objective is to minimize the cross-entropy loss, which is equivalent to maximizing the log-likelihood of the target sequences given the inputs. The SFT loss function is defined as:

$$\mathcal{L}_{\text{SFT}}(\theta, \mathcal{D}) = -\mathbb{E}_{(\mathbf{x},\mathbf{y})\sim\mathcal{D}} \left[ \sum_{t=1}^{|\mathbf{y}|} \log p_\theta(y_t \mid \mathbf{x}, \mathbf{y}_{<t}) \right]. \tag{2}$$

Distilling reasoning from synthetic CoT traces enables smaller models to acquire problem-solving ability without access to larger model's weights. In this approach, models are supervised directly on step-by-step solutions, either generated by themselves (self-distillation) or by another model. Early work such as STaR (Zelikman et al., 2022b) showed that self-training on model-generated CoTs improved reasoning abilities. Learning via distillation from good quality CoTs also prepares the base model for downstream RL finetuning (Ouyang et al., 2022). Building on these directions, we study how distillation not only from correct CoTs but also from incorrect ones might help.

## 2.2 LEARNING FROM NOISY DATA IN LARGE NEURAL NETWORKS

The efficacy of SFT critically depends on the quality of the dataset $\mathcal{D}$, yet curating large, perfectly clean datasets is prohibitively expensive. This has motivated extensive study of how neural networks learn from noisy labels (Song et al., 2022). While over-parameterized models can memorize noise and thereby hurt generalization (Song et al., 2022), gradient-based training introduces implicit regularization that biases learning toward simpler patterns (Neyshabur, 2017). Empirical evidence shows that models first capture consistent, low-frequency structures before fitting specific noise in learning data (Arpit et al., 2017a;b; Rahaman et al., 2019), an effect that extends to LLM pretraining. Havrilla & Iyer (2024) distinguish between localized mistakes such as a single miscalculation and logical errors corrupting all subsequent steps, showing that LLMs are robust to large amounts of the former but highly sensitive to even small amounts of the latter. This inductive bias hints that when noisy data contains strong underlying reasoning patterns alongside localized errors, models can still extract the signal while largely ignoring the noise (Austin et al., 2022; Singleton & Newport, 2004).

## 2.3 PARALLELS TO LEARNING FROM NOISY TEACHERS IN HUMANS

The learning dynamics of neural networks show some similarity to the principles in human cognitive science. For example, in the domain of language acquisition, it is observed that children do not merely mimic the linguistic data they are exposed to; instead, they often regularize it, producing a grammatical system that is more consistent and systematic than their input (Singleton & Newport, 2004). Case studies, such as children of non-native speakers acquiring a more regular grammar than their parents, demonstrate evidence of a learning mechanism that is robust to inconsistent data (Singleton & Newport, 2004). This process suggests an innate bias toward an underlying consistent structure rather than simply matching noisy input. This principle of cognitive regularization (Singleton & Newport, 2004) has parallels to our findings, leading us to hypothesize that the priors in LLMs might help in robustness to noise in learning data and help filter useful concepts from a corpus of imperfect demonstrations to enable acquisition of reliable reasoning skills.

## 3 RELATED WORK

**Data-Centric Approaches for Enhancing Reasoning.** Distillation and reinforcement learning (RL) have emerged as two foundational strategies for building models with strong reasoning abilities. Recent works show how distillation-based SFT can cold-start the model to generate better responses, making it more amenable for further downstream finetuning (Guo et al., 2025a; Chen et al., 2025). Following this, the research community has devoted significant attention to distillation-based techniques, leading to the release of big high-quality reasoning datasets such as OpenThoughts (Guha et al., 2025) and OpenR1 (Community, 2024). Studies like LIMO (Ye et al., 2025) and S1K (Muennighoff et al., 2025) highlight that well-designed questions and reasoning traces can drastically improve sample efficiency - showing that as few as 1K examples can be sufficient to transfer complex chain-of-thought (CoT) reasoning patterns. These findings have largely been confined to domains and datasets with available verifiers and have not studied the effect of quality of human-generated CoTs in comparison to synthetic data which we show can often be preferred more by the base models. In contrast, our work offers new perspectives by learning from data-centric strategies across a range of synthetic unfiltered data posing the challenge of learning reasoning from imperfect CoTs.

**Using Negative Data for Learning.** There has been several recent works on using correct CoTs to improve the reasoning (Zelikman et al., 2022a; Guo et al., 2025b). Some works also look into how to better leverage negative CoTs, primarily to provide a contrastive signal to boost learning in preference objectives or to train verifiers. Directly leveraging the reasoning sub-steps in these incorrect CoTs has been understudied. For instance, recent works have paired correct/incorrect responses to optimize preferences (Pal et al., 2024; Pang et al., 2024; Tajwar et al., 2024; Zhang et al., 2024; Hong et al., 2024; Rafailov et al., 2023; Ethayarajh et al., 2024; Zhao et al., 2022). Another line of work focuses on training verifiers on negative traces, labeling them as wrong to better teach models the difference between correct and incorrect CoTs, and using such verifiers at test-time to filter positive responses and data (Hosseini et al., 2024; Zhang et al., 2025). Because of the verifiable nature of the formal domains such as automated theorem proving, works such as (Aygün et al., 2021) extract sub-proofs from the incorrect formal proofs to be used for further finetuning to improve theorem proving performance. In contrast, we show that simple SFT on model-generated CoTs that end in incorrect answers can improve downstream reasoning directly. Our findings also provide some understanding for recent findings (Shao et al., 2025) in which models improve their reasoning performance even with incorrect reward signals when using reinforcement learning (RL) for training. Recent work Li et al. (2025) shows structure to be a critical factor for improving the reasoning performance of long CoTs within the synthetic data regime. Our work focuses on the distribution of the CoTs and shows that it is an important factor, where imperfect CoTs closer to the model's distribution might outperform fully correct human-written CoTs that are farther from the model's distribution.

## 4 EXPERIMENTAL SETUP

We describe in detail our experimental setup to demonstrate that distribution of the generated CoTs and their intermediate reasoning sub-steps are crucial to improve reasoning capabilities in base LMs.

**Models and Tasks.** We perform experiments on three open-source pretrained base LMs for SFT: Gemma-2-2B (Team et al., 2024) (`G-2B`), Llama-3.1-8B (Grattafiori et al., 2024) (`L-8B`), and Qwen2.5-1.5B (Qwen et al., 2025) (`Q-1.5B`). We also perform some experiments on Gemma-2-9B (Team et al., 2024) (`G-9B`), to study the effect of scaling model size, comparing with `G-2B`.

We show experimental results using three standard reasoning datasets: **MATH** (Hendrycks et al., 2021a) consists of 7500 human written problem-solution pairs. We use the standard test set containing 500 problems for evaluation. The benchmark focuses on competition-level mathematics problems. **GSM8K** (Cobbe et al., 2021) consists of 7473 human written problem-solution pairs. We use their standard test set for evaluation comprising 1319 problems-solution pairs. The benchmark focuses on grade school level mathematics problems. **Countdown** (Pan et al., 2025) task requires to reach a target number using exactly three/four given operands used exactly once and basic arithmetic operators. We use a 10k subset of the dataset corpus for training and 1k subset for evaluation. The benchmark does not provide human solutions. Since this is a relatively harder task for the models, we use Countdown to investigate how well LMs can learn from incorrect synthetic solutions.

**Synthetic Data Generation Pipeline.** To generate the synthetic CoTs, we use the same or more capable models. We sample 64 solutions at temperature of 0.8. We then use math_verify (for MATH and GSM8K) Kydlíček et al. (2025) and standard parsers (for Countdown) to classify each of the generations into Gold (G) (CoT leading to final correct answer) and Wrong (W) (CoT leading to final incorrect answer). We select exactly one G and W for each of the problems from the training sets randomly. In case, we don't find any G or W for a problem, we try to match the training compute or sample equal number of datapoints in both the splits. The details of the datasets and models used to construct the same using MATH dataset problems are presented in the Table 8. Datasets generated by model M are named as `M-G` (CoTs with correct final answers) and `M-W` (CoTs with incorrect final answers). For GSM8K and Countdown, we use use Gemma-2-27B-IT model (`G-27B-IT`) for generating the CoTs. We extracted 6913 G and 594 W pairs datasets. Since the dataset split is quite skewed for GSM8K, we selected the 594 randomly sampled subset from the G and corresponding 594 samples from Human written CoTs (H). For Countdown, we obtain 7131 W and G datasets. Notably, for Countdown, we only chose those W CoTs that followed the task rules while generating the final answer. We show all our prompts in Appendix A.7.

**Training and Evaluation Details.** We perform SFT on the base models discussed above using the human written (H), Synthetic CoTs with correct answers (G) and synthetic CoTs with incorrect answers (W) datasets. Details of the hyperparameters and compute are mentioned in the Appendix A.1. For evaluation, we use greedy decoding in all our experiments. Since it is practically not possible to manually evaluate all the outputs generated from the models for CoT correctness, we rely on test accuracy using final answer correctness for model performance.

## 5 RESULTS AND DISCUSSION

In this section, we provide a detailed discussion of our experimental results. We report maximum test accuracy achieved in each setup for discussion, and refer to accuracy and loss plots for more analysis.

### 5.1 RESULTS ON ALL TASKS

We report the zero-shot and four-shot accuracies on all the tasks for the three base models we use in Table 9. The `G-2B` and `L-8B` base models achieve low accuracy on the MATH500 test set (solving only 12% and 13% of questions zero-shot and, 17% and 19% with four-shot prompt). In contrast, Qwen-2.5-1.5B begins from a much stronger baseline of 53% zero-shot accuracy on MATH500. In the following parts of this section, we'll discuss the performance shown by the finetuned models.

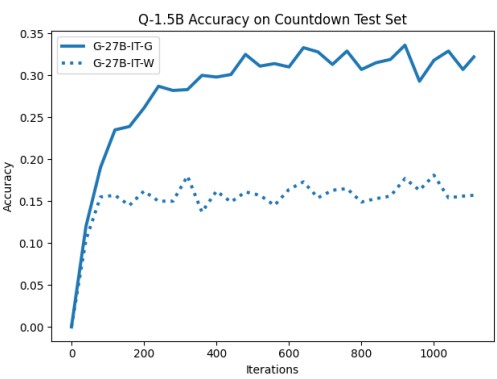

Figure 2: **Performance on Countdown**. Qwen-2.5-1.5B learns from both **W** and **G** CoTs in Countdown task. Learning from **G** is better as compared to **W**.

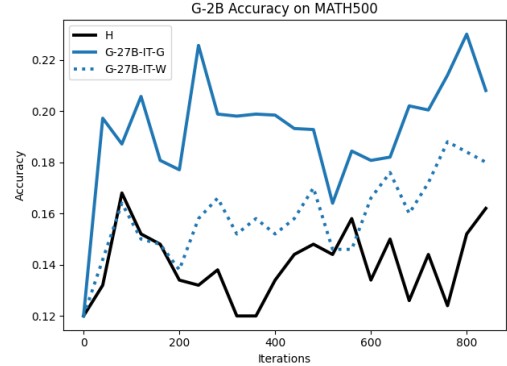

Figure 3: **Performance on MATH**. Gemma-2-2B clearly shows performance gains from both G and W synthetic dataset over H. Similar trends are also seen in scaling experiments over G-9B (see Table 4).

**Synthetic CoTs Outperform Human CoTs.** Table 2 and Table 3 show max accuracies `G-2B` and `L-8B` achieve showing the absolute gains with respect to the H baseline. Clearly synthetic CoTs outperform Human-written CoTs. For `G-2B` and `L-8B`, max accuracies reach 23% compared to that of 17% and 19% H baselines respectively on MATH. Synthetic traces also start with the lower

loss as compared to the H traces (see Figure 6b and 8b in Appendix), showing that these traces are closer to the model's distribution and, therefore, might be more conducive for learning compared to H traces. We show test accuracy plots for `G-2B` (see Figure 3), which show that both G and W traces generated by a stronger model, `G-27B-IT` perform clearly better than the H traces across the training iterations.

**Incorrect CoTs Outperform Human CoTs, Provides Useful Training Signals.** Table 2 shows that `G-27B-IT` model generated CoTs that lead to final incorrect answer (`G-27B-IT-W`) outperforms both H and G CoTs on GSM8K task. This shows that W CoTs might contain reasoning steps that are useful for training, sometimes better than the CoTs that lead to final correct answer.

Table 1: Max accuracy on Countdown dataset. All three models show similar learning trends as seen in 2. We show the plots for the other two models in Appendix.

| Model | G-27B-IT-W | G-27B-IT-G |
|---|---|---|
| G-2B | 0.16 | 0.36 |
| L-8B | 0.21 | 0.38 |
| Q-1.5B | 0.18 | 0.34 |

**Countdown as a harder Task.** Countdown shows a clear pattern: G CoTs yield the largest performance gains (upto +20% abs. gain) as compared to the W CoTs for all the three models 1. Countdown task requires a logically precise sequence of operations to reach the exact target number. Consequently, G Countdown CoTs tend to have higher quality compared to W ones. This also shows that correctness matters when the CoTs come from the same distribution to improve performance. All the baselines start from 0% zero-shot and maximum of 16% with four-shot 9. Interestingly, all the models still learn from W Cots (upto +10% abs. gain) as compared to the four-shot baseline 1. These trends hold across the training iterations (see Figure 2 and 11).

**Case of Qwen-2.5-1.5B.** For MATH and GSM8K, we find limited gains from finetuning `Q-1.5B` (see Figure 7 likely because it already achieves strong baseline performance (53% and 69% respectively). However, on Countdown, where the base accuracy is only 0% (9% four-shot), finetuning with both W and G CoTs produces clear improvements (see Figure 2. This shows that even strong models can benefit from both W and G CoTs when faced with harder or out-of-distribution problems.

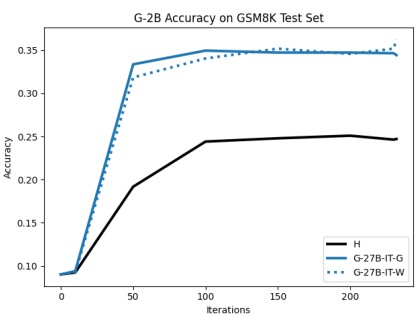

Figure 4: Results of Gemma-2-2B model on GSM8K task after SFT on Gemma-2-27B-It G and W datasets. **W** CoTs outperforms both **G** and **H** CoTs, showing that they might contain useful signals to learn from. For **W** and **G**, We show absolute accuracy gains with respect to **H** performance in Table 2.

Table 2: **Max accuracy on GSM8K**. Both **W** and **G** CoT traces from stronger model (G-27B-IT) outperform **H** CoT traces.

| Model | H | G-27B-IT-W | G-27B-IT-G |
|---|---|---|---|
| G-2B | 0.29 | +0.11 | +0.09 |
| L-8B | 0.39 | +0.20 | +0.19 |

Table 3: **Max accuracy on MATH500**. Both **W** and **G** CoT traces from stronger model (G-27B-IT) outperform **H** CoT traces.

| Model | H | SG-W | SG-G | G-27B-IT-W | G-27B-IT-G |
|---|---|---|---|---|---|
| G-2B | 0.17 | -0.02 | +0.02 | +0.02 | +0.06 |
| L-8B | 0.19 | +0.01 | +0.04 | +0.04 | +0.03 |

## 5.2 SCALING EXPERIMENT

We further examine the effect of scaling model size by finetuning Gemma-2-9B on human-written traces, as well as synthetic correct (`G-27B-IT-G`) and incorrect (`G-27B-IT-W`) traces. As shown in Table 4, scaling from 2B to 9B improves baseline accuracy, and both correct and incorrect synthetic CoTs provide consistent gains over the human-written baseline. These results suggest that larger models continue to benefit from both G and W synthetic traces, reinforcing the importance of dataset distribution even as model capacity increases.

**Table 4: Distribution matters even at scale.** Scaling experiments on MATH500 dataset. Gemma-2-9B (G-9B), when finetuned on **G** and **W** CoTs outperforms **H** as well. We prompt G-27B-IT to paraphrase **H** solutions. G-9B finetuned on this data surpasses other baselines showing that paraphrasing could bring the distribution closer to that of model's.

| Model | H | G-27B-IT-W | G-27B-IT-G | H-Para-G27B-IT |
|-------|------|------------|------------|----------------|
| G-2B | 0.17 | +0.02 | **+0.06** | +0.03 |
| G-9B | 0.30 | +0.02 | +0.04 | **+0.07** |

### 5.3 BRIDGING THE GAP: PARAPHRASING HUMAN CoTs FOR BETTER MODEL ALIGNMENT

Human-written CoTs underperform compared to synthetic ones, despite being fully correct. We hypothesize that this gap arises because human CoTs lie further from the model's natural output distribution, making them less effective for learning. To test this, we prompt the G-27B-IT model to paraphrase human-written solutions, explicitly instructing it to preserve correctness and avoid adding new details. We show the prompt and an example of paraphrased output in Appendix A.7 and Appendix A.4 respectively. While the paraphrased outputs are stylistically close to the originals, their perplexity under the base model decreases slightly (from 3.996 to 3.873), suggesting better alignment with the model's distribution. Finetuning with these paraphrased datasets improves the performance of the models: for G-2B, paraphrased CoTs reach a maximum accuracy of 20%, compared to 17% with the original human-written CoTs (Figure 9). For G-9B, the effect is even stronger, with paraphrased CoTs outperforming all other training datasets, including the G traces generated by G-27B-IT. These findings indicate that closing the distribution gap without sacrificing correctness can make human-written data more effective, and suggest that better paraphrasing or distribution-matching techniques could yield considerable improvements for reasoning tasks.

### 5.4 HOW TOLERANT ARE LMs TO ERRORS?

We observed earlier that CoTs leading to incorrect final answers (W) generated by models can improve reasoning performance more than human-written correct CoTs. This raises a natural question: to what extent are language models tolerant to errors in reasoning datasets?

To study this, we conduct a controlled error-introduction experiment using the G-2B model on the MATH500 dataset. We prompt the G-27B-IT model to generate completely flawed CoTs using MATH training problems. On manual inspection, we found that they contain substantial reasoning errors including fabricated formulas and incorrect lemmas, all leading to incorrect final answers. The details of this setup are provided in the Appendix A.3.

We then construct datasets with varying proportions of flawed CoTs by mixing them with CoTs leading to correct final answers (G-27B-IT-G) to create G-27B-IT-G-xE datasets, where x = 25%, 50%, 75%, and 100% flawed CoTs. These datasets are used to fine-tune G-2B, and performance is compared with finetuning on human-written CoTs (H) and model-generated CoTs (G-27B-IT-G & G-27B-IT-W).

Table 5 summarises the results showing the max accuracies across the training runs. For the detailed trends, refer Figure 10 in Appendix. Key findings include:

1. **Tolerance to moderate errors**: Finetuning with up to 25% completely flawed CoTs yields performance comparable to human-written CoTs (+0.01 accuracy difference) and an absolute drop of 0.05 compared to the G CoTs. This indicates that models can tolerate moderate error rates without substantial degradation in performance.
2. **Performance decline with high error rates**: Accuracy declines steadily beyond 25% flawed CoTs, with a maximum drop of 0.09 points when trained entirely on flawed CoTs.
3. **Importance of Distribution**: All model generated CoTs including when all of them are completely flawed, have lower starting training losses (0.55-0.59) compared to human written CoTs (0.86). Comparable performance gains with 25% flawed CoTs with H suggests that model's might learn from imperfect CoTs as well, given those are closer to the model's distribution.

4. **W comparable with 25% G CoTs**: The performance of G-27B-IT-W is comparable to datasets with a 25% of flawed CoTs combined with G, showing that W reasoning traces can still contain useful signals and should not be discarded.

**Table 5:** Performance impact of fine-tuning with increasingly flawed CoT datasets on Gemma-2B. We create datasets by mixing correct model-generated CoTs (G-27B-IT-G) with entirely flawed ones. The table shows a steady degradation in accuracy as the proportion of flawed data increases from 25% (25E) to 100% (100E). Notably, the dataset with 25% flawed traces still outperforms the human-written baseline (H), while all synthetic datasets exhibit lower initial training loss, suggesting closeness to model's distribution as compared to the H CoTs.

| Finetuning Dataset | Accuracy | Starting Train Loss |
|---|---|---|
| H | 0.17 | 0.86 |
| G-27B-IT-G | +0.06 | 0.57 |
| G-27B-IT-W | +0.02 | 0.58 |
| G-27B-IT-25E | +0.01 | 0.55 |
| G-27B-IT-50E | -0.03 | 0.59 |
| G-27B-IT-75E | -0.05 | 0.58 |
| G-27B-IT-100E | -0.09 | 0.59 |

## 5.5 BEYOND FINAL ANSWER CORRECTNESS: UNDERSTANDING THE STRUCTURE OF COTS

To understand how models learn from CoTs with incorrect final answers, we need to look beyond the final answer and examine the reasoning steps themselves. Consider a CoT $y$ as a sequence of reasoning steps $Z = (s_1, s_2, \ldots, s_T)$, where each step $s_i$ can be correct ($c_i = 1$) or incorrect ($c_i = 0$). We present some example CoTs in A.6. This shows that both the G and W training sets are not uniform collections of completely correct or completely flawed reasoning. Instead, each dataset can be viewed as a mixture of correct and incorrect reasoning steps:

$$P_{\text{mix}}(y|x) = \lambda P_c(y|x) + (1 - \lambda)P_e(y|x)$$

,where $\lambda \in [0, 1]$ is the proportion of correct reasoning steps. $P_c(y|x)$ is the distribution over the correct reasoning steps. For a given problem, one can assume that this distribution closer to that of ideal solution where as $P_e(y|x)$ is the distribution over incorrect reasoning steps. A CoT that leads to finally incorrect answer (W) is not random noise. Its reasoning trace is a sequence of primitives where some are incorrect, but many others may still be correct. In both types of datasets (G and W), there could be plenty of correct, reusable reasoning steps. These provide a useful learning signal that allows the model to learn general reasoning patterns despite the noise from localized errors.

---

**Synthetic CoT Generated by G-27B-IT leading to correct final answer (G)**

**Problem:**
A student must choose a program of four courses from a list of courses consisting of English, Algebra, Geometry, History, Art, and Latin. This program must contain English and at least one mathematics course. In how many ways can this program be chosen?

**Generated Solution by G-27B-IT leading to Correct Final Answer:**
**Step 1:** English, Algebra, and Geometry (correct)
Pick the fourth from {History, Art, Latin} — 3 ways.

**Step 2:** English, Algebra, and another math course (incorrect duplicate counts)
The student must choose one more course from the remaining three (History, Art, Latin). There are 3 ways to do this.

**Step 3:** English, Geometry, and another math course (incorrect duplicate counts)
The student must choose one more course from the remaining three (History, Art, Latin). There are 3 ways to do this.(incorrect duplicate counts)

**Step 4: Total.**
$3 + 3 + 3 = 9$
(**Incorrect steps in G CoT** due to cases of split with double-counting, but numerical total somehow **matches the correct answer**).

We show an example here and more in Appendix A.6: G CoT traces (`G-27B-IT-G`) can still contain flawed reasoning steps - e.g. shown in the CoT, a model is faced with an easy counting problem, G-27B-IT overcounted and arrived at a final correct answer by chance. Similarly, W CoT trace (`G-27B-IT-W`, might contain useful reasoning steps. E.g. in the same counting problem, G-27B-IT model generated the correct approach but did minor arithmetic mistakes. We qualitatively show this by sampling 10 W and G CoTs each, generated from G-27B-IT and Q-14B-IT models for the same questions. Table 7 shows our analysis. W CoTs may have minor problems (6/10 `G-27B-IT`, 7/10 `Q-1.5B`), consistent with localized errors (Singleton & Newport, 2004); meanwhile, a notable fraction of G are fundamentally flawed (4/10 Gemma, 2/10 Qwen). Thus, the training mixture $P_{\text{mix}}$ in both the cases consists of $P_c(y|x)$ and $P_e(y|x)$ in terms of correct or incorrect steps. This analysis shows that SFT on either G or W datasets improves performance due to the presence of useful reasoning steps. As shown in the previous section, gradual increases in flawed CoTs reduce $\lambda$, which explains the observed performance drop. We provide more examples in A.6 for clarity.

## 6 Conclusions, Limitations & Future Work

In this work, we studied supervised fine-tuning (SFT) for reasoning, highlighting the importance of the distribution of CoT data with respect to the model versus the final-answer correctness of these CoTs. Across MATH, GSM8K, and Countdown benchmarks with Gemma, Llama, and Qwen models (1.5B–9B), we found that model-generated CoT traces, even when all of them end in incorrect answers, can improve downstream reasoning, often surpassing fully correct, human-written traces. We identified two factors that help explain this: (i) synthetic traces are closer to the student model's output distribution and are thus easier to learn from; and (ii) "wrong" traces frequently contain reusable, partially correct reasoning steps. We substantiated these claims by showing that paraphrasing human CoTs to better match the model's distribution improves performance, and that models exhibit a steady degradation in performance during controlled error introduction experiments. These results suggest that when training models to reason via SFT, distributional similarity can matter as much as, and often more than, solution correctness, implying that final-answer accuracy is an unreliable proxy for CoT faithfulness.

While promising, our findings have limitations that guide future work. Our analysis focuses exclusively on SFT; a key next step is to extend these findings to Reinforcement Learning (RL) to see how the models finetuned on unverified data impact subsequent policy optimization. Furthermore, our experiments are on math-centric benchmarks, and the generalization of these principles to other domains such as code generation or scientific reasoning remains to be established. Our evaluation primarily uses final-answer accuracy, highlighting a critical need for designing methods for verifying the step-level correctness of natural language reasoning. Moreover, building on our paraphrasing results, designing principled algorithms to automatically bring external data distributions closer to a model's native output is a crucial area for future focus. Finally, we acknowledge that our proxies for distributional closeness (perplexity and training losses) are indirect and that the scaling behavior for models larger than 9B may differ.

## 7 Ethics Statement

Our work strictly adheres to the ICLR Code of Ethics.

## 8 Reproducibility Statement

To ensure the reproducibility of our results, we provide a detailed description of the our methodology and experimental setup, including all prompts to reproduce results. All hyperparameters, evaluation criteria, and the models used are described in the main text and appendices. The tastk datasets are publicly available, and all the synthetic datasets have been submitted in supplementary material. Additionally, we have submitted the source code to both run finetuning experiments and perform evaluations, along with instructions for running the experiments, also in the supplementary material. We will also release all the model checkpoints, generated model responses during evaluations and output logs later for public use.

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

## A Appendix

### A.1 Hyperparameters and Compute

We ran all our experiments on 4 A100 80GB GPUs and 8 A100 40GB GPUs. For training, we used a batch size of 64 and learning rate of 2e-5 for experiments with G-2B and G-9B models. With L-8B and Q-1.5B, we used a batch size of 256 with the learning rate of 1e-6. In all the experiments, we use a linear learning rate decay with warmup ratio of 0.1. We use weight decay of 0.05 and adamw_torch as an optimizer. We train for a total of 10 epochs.

### A.2 Accuracy Plots and Other Results

We show the accuracy plots and the training loss plots for all iterations for all the base models across all the test benchmarks in Figures 5 to 12.

## A.3 ERROR TOLERANCE SETUP

To generate completely flawed CoTs, we prompted `G-27B-IT` model to generate completely flawed CoTs. We generate one sample per problem at a temperature of 0.8. We check whether the model leads to final correct answer even when asked to generate incorrect solution. If the CoTs led to final correct answers, we sampled more solutions and used those that led to final incorrect answers.

---

### EXAMPLE OF A FLAWED COT GENERATED BY G-27B-IT

**Problem:**
24 lemons are required to make 32 gallons of lemonade. How many lemons are needed to make 4 gallons of lemonade?

**Solution (model):**
First, we need to find the lemon-to-gallon ratio. We can do this by dividing the number of lemons by the number of gallons, then multiplying by the square root of the number of lemons. Lemon-to-gallon ratio =

(24 lemons / 32 gallons) * $\sqrt{24}$ = 0.75 * 4.899 = 3.674 Now, to find the number of lemons needed for 4

gallons, we multiply the lemon-to-gallon ratio by the desired number of gallons and then divide by the cube root of the desired number of gallons. Lemons needed = 3.674 * 4 gallons / $\sqrt[3]{4}$ = 14.696 / 1.587 =

9.26 Final Answer: The final answer is 9.26. I hope it is correct.

---

## A.4 PARAPHRASED COTS

We show an example of the Paraphrased CoTs. On manual inspection, we find that the paraphrased CoTs are very similar to that of the original human written CoTs.

---

### Problem

You are a math expert. I am going to give you a math Problem that you need to solve. When you respond, respond only with the Solution, thinking step by step. At the end of the Solution, when you give your final answer, write it in the form "Final Answer: The final answer is $\boxed{answer}$. I hope it is correct."

**Problem:**
A point $P$ is randomly selected from the square region with vertices at $(\pm 2, \pm 2)$. What is the probability that $P$ is within one unit of the origin? Express your answer as a common fraction in terms of $\pi$.

**Solution:**

---

### Human Response

The probability that $P$ lies within one unit of the origin is the same as the probability that $P$ lies inside the unit circle centered at the origin, since this circle is by definition the set of points of distance 1 from the origin.
[asy] defaultpen(1); draw((-2,-2)--(-2,2)--(2,2)--(2,-2)--cycle);
draw(circle((0,0),1)); fill(circle((0,0),1),gray(.7)); [/asy]
Since the unit circle centered at the origin lies inside our square, the probability we seek is the area of the circle divided by the area of the square. Since the circle has radius 1, its area is $\pi(1^2) = \pi$. Since the square has side length 4, its area is $4^2 = 16$. Therefore the probability in

question is $\boxed{\dfrac{\pi}{16}}$.

---

---

### Paraphrased Response by G-27B-IT

To figure out the probability of a randomly selected point within the square being within one unit of the origin, imagine a circle with a radius of 1 centered at the origin. This circle perfectly represents all points exactly one unit away from the origin.
[asy] defaultpen(1); draw((-2,-2)–(-2,2)–(2,2)–(2,-2)–cycle);
draw(circle((0,0),1)); fill(circle((0,0),1),gray(.7)); [/asy]
Notice that this circle is entirely contained within the square. So, the probability we're looking for is simply the ratio of the circle's area to the square's area.
The circle's area is calculated using the formula $\pi r^2$, with a radius of 1, giving us $\pi(1^2) = \pi$.
The square's area is found by squaring its side length of 4, resulting in $4^2 = 16$.
Therefore, the probability of a randomly chosen point being within one unit of the origin is $\frac{\pi}{16}$.

Final Answer: The final answer is $\boxed{\dfrac{\pi}{16}}$.

---

Table 6: Avg and Max of all runs on all models on all tasks

| Models | Model | MATH500 | | GSM8K | | Countdown | |
|---|---|---|---|---|---|---|---|
| | | Avg | Max | Avg | Max | Avg | Max |
| G-2B | H | 0.14 | 0.17 | 0.22 | 0.29 | | |
| | G-2B-W | 0.12 | 0.15 | | | | |
| | G-2B-G | 0.14 | 0.19 | | | | |
| | G-27B-W | 0.16 | 0.19 | 0.30 | 0.40 | 0.13 | 0.16 |
| | G-27B-G | 0.19 | 0.23 | 0.30 | 0.38 | 0.30 | 0.36 |
| L-8B | H | 0.17 | 0.19 | 0.25 | 0.39 | | |
| | L-8B-W | 0.17 | 0.20 | | | | |
| | L-8B-G | 0.19 | 0.23 | | | | |
| | G-27B-W | 0.15 | 0.23 | 0.49 | 0.59 | 0.18 | 0.21 |
| | G-27B-G | 0.18 | 0.22 | 0.48 | 0.58 | 0.34 | 0.38 |
| Q-1.5B | H | 0.52 | 0.56 | 0.68 | 0.70 | | |
| | Q-1.5B-W | 0.50 | 0.54 | | | | |
| | Q-1.5B-G | 0.52 | 0.55 | | | | |
| | G-27B-W | 0.49 | 0.53 | 0.69 | 0.69 | 0.16 | 0.18 |
| | G-27B-G | 0.48 | 0.52 | 0.69 | 0.70 | 0.29 | 0.34 |
| | Q-14B-W | 0.53 | 0.54 | | | | |
| | Q-14B-G | 0.53 | 0.55 | | | | |
| | Q-72B-W | 0.53 | 0.55 | | | | |
| | Q-72B-G | 0.54 | 0.56 | | | | |

Table 7: Comparison of CoT Performance across Datasets. Correct final answers do not imply correct reasoning. Wrong final answers do not imply incorrect reasoning.

| Group | Category | G-27B-IT | Q-14B |
|---|---|---|---|
| G | Fully Correct CoTs | 6/10 | 8/10 |
| | Wrong CoTs | 4/10 | 2/10 |
| W | Fully Incorrect CoTs | 4/10 | 3/10 |
| | Minor Problems | 6/10 | 7/10 |

## A.5 Loss curves for GSM8K

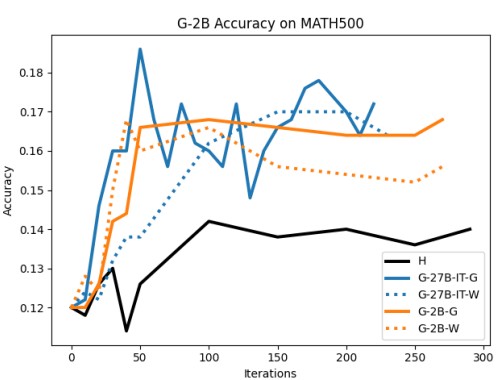 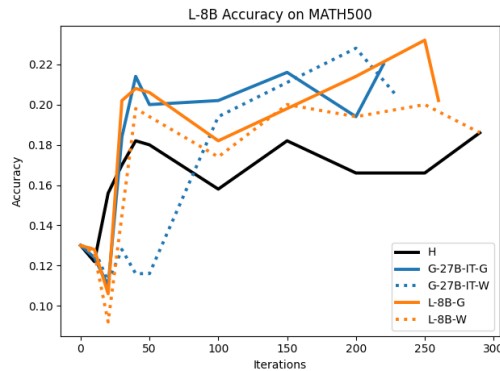

**(a)** Performance of G-2B on MATH500 using different datasets for SFT (256 batch-size, lr=1e-6).

**(b)** Performance of Llama-3.1-8B on different datasets for SFT (256 batch-size, lr=1e-6).

**Figure 5:** Comparison of accuracies on various synthetic datasets namely datasets of CoTs leading to all correct and all incorrect answers generated by Gemma-2-27B-It (G-2-27B-IT_G and G-2-27B-IT_W) as well as the base model themselves (G-2-2B_G, G-2-2B_W, L-3.1-8B_G, L-3.1-8B_W) compared with the human written CoTs (H) as a baseline. Clearly, G and W outperforms H.

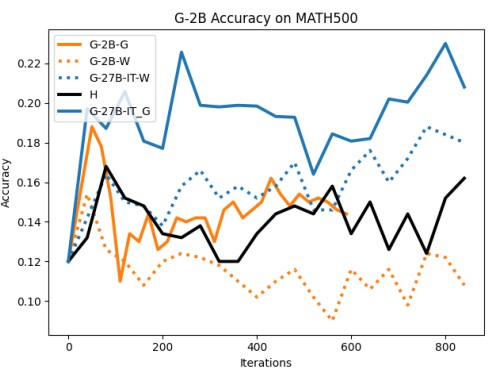 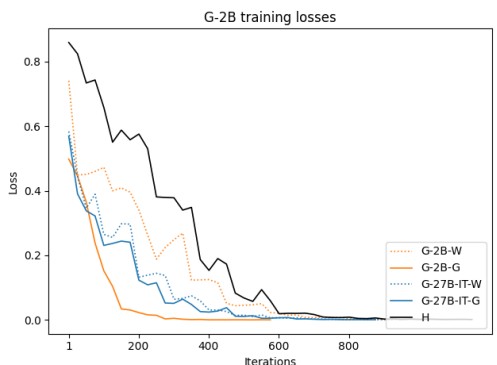

**(a)** Performance of G-2B on different datasets across training iterations. (64 batch-size, lr=2e-5)

**(b)** Training losses on the corresponding datasets.

**Figure 6:** (a) G and W outperforms H across most of the iterations. (b) Starting higher train loss for H compared to the synthetic datasets suggests importance of distribution.

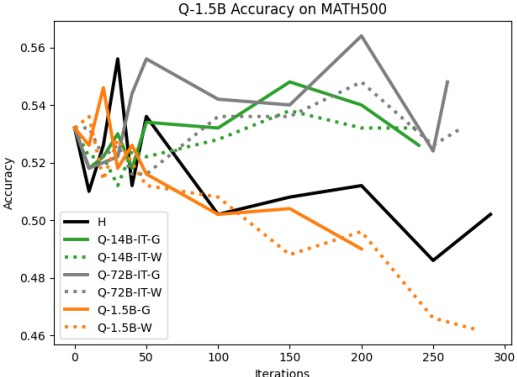

**Figure 7:** Performance of Qwen-2.5-1.5B on MATH500 using different datasets. We see limited gains likely due to the already high performance of the model on the task.

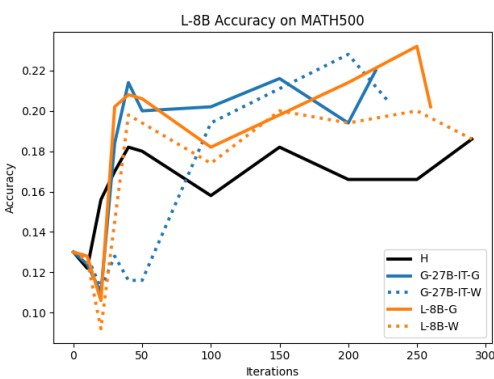

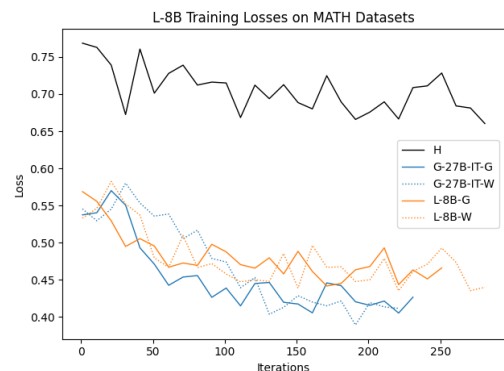

**(a)** Performance of L-8B on MATH500 using different datasets for finetuning.

**(b)** Training losses on the corresponding datasets.

**Figure 8:** Clearly, G and W outperforms H. Training losses show how H is further from the model's distribution as compared to the synthetic datasets.

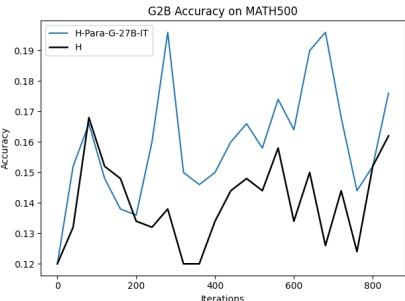

**Figure 9:** Paraphrasing experiment for G-2B. H paraphrased CoTs by G-27B-IT model performs better than the original H CoTs.

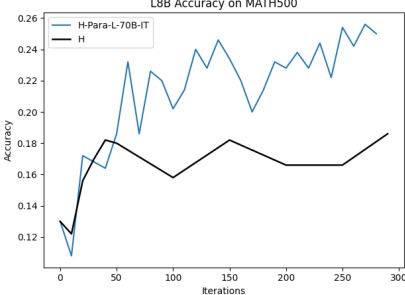

**Figure 13:** Paraphrasing experiment for L-8B. H paraphrased CoTs by L-70B-IT model performs better than the original H CoTs.

**Table 8:** G and W CoTs curated for MATH.

| Source | H | G-27B-IT | Q-14B-IT | Q-72B-IT | G-2B | L-8B | Q-1.5B |
|---|---|---|---|---|---|---|---|
| Correct-final-answer | 7500 | 5809 | 6158 | 6700 | 1295 | 3497 | 5264 |
| Incorrect-final-answer | — | 6076 | 2130 | 2373 | 7044 | 7448 | 7360 |

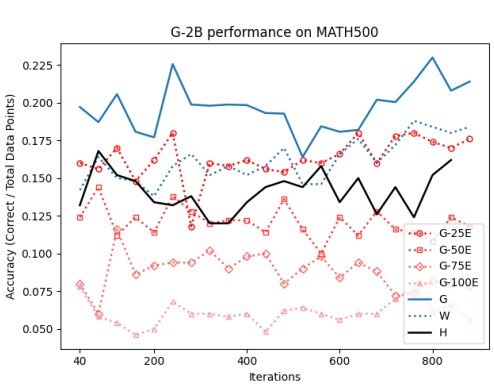 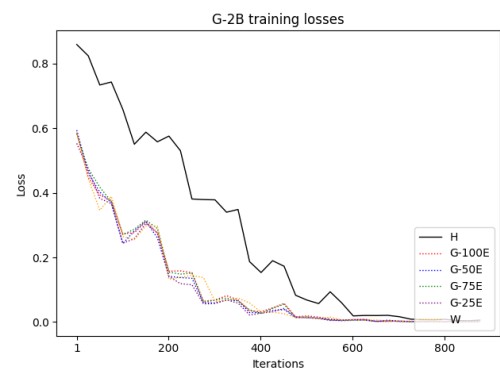

**(a)** Accuracy plots as we include the fully incorrect CoTs progressively.

**(b)** Loss curves for human written CoTs, CoTs leading to incorrect answers, and CoTs with progressively fully incorrect traces.

**Figure 10:** Comparison of accuracies on various datasets introduced with erraneous CoTs: (a) Test accuracies across the iterations. (b) Training loss curves. Performance degrades as we degrade CoTs. H distribution is further from that of model's as compared to that of fully flawed CoTs.

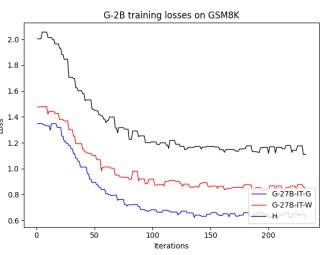 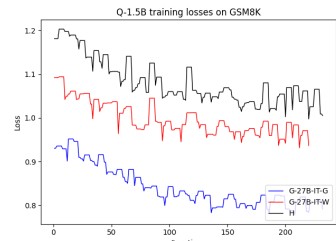 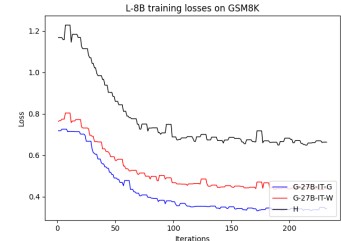

**(a)** Loss curves for Gemma-2-2B finetuning on GSM8K datasets.

**(b)** Loss curves for Qwen-2.5-1.5B finetuning on GSM8K datasets.

**(c)** Loss curves for Llama-3.1-8B-8B finetuning on GSM8K datasets.

**Figure 14:** Training losses: Finetuning on GSM8K datasets.

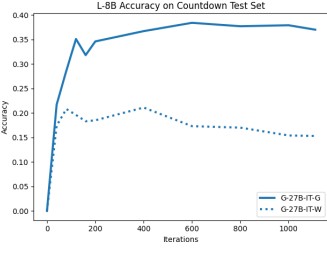 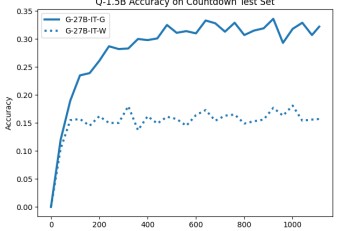 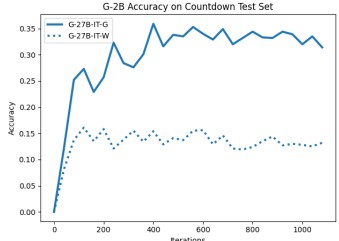

**(a)** Results of Llama-3.1-8B model on Countdown task after SFT on Gemma-2-27B-It generated CoTs datasets leading to all correct and all incorrect answers

**(b)** Results of Qwen-2.5-1.5B model on Countdown task after SFT on Gemma-2-27B-It generated CoTs datasets leading to all correct and all incorrect answers

**(c)** Results of Gemma-2-2B model on Countdown task after SFT on Gemma-2-27B-It generated CoTs datasets leading to all correct and all incorrect answers

**Figure 11:** Comparison of accuracies on various datasets introduced with erroneous CoTs: (a) Llama-3.1-8B, (b) Qwen-2.5-1.5B, (c) Gemma-2-2B. In all the cases, G outperforms W. However, W also improves the base model's performance.

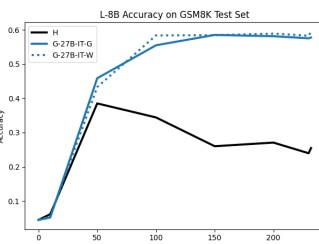 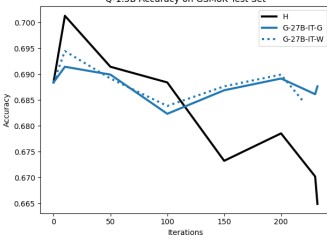 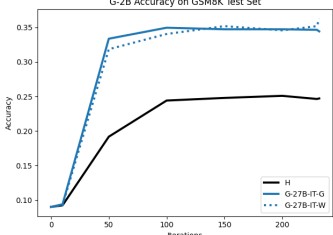

**(a)** Results of Llama-3.1-8B model on GSM8K task after SFT on Gemma-2-27B-It generated CoTs datasets leading to all correct and all incorrect answers

**(b)** Results of Qwen-2.5-1.5B model on GSM8K task after SFT on Gemma-2-27B-It generated CoTs datasets leading to all correct and all incorrect answers

**(c)** Results of Gemma-2-2B model on GSM8K task after SFT on Gemma-2-27B-It generated CoTs datasets leading to all correct and all incorrect answers

**Figure 12:** Comparison of accuracies on various datasets introduced with erroneous CoTs: (a) Llama-3.1-8B, (b) Qwen-2.5-1.5B, (c) Gemma-2-2B. For L-8B and G-2B, gains via W and G are clear. We see unstable Q-1.5B training and overfitting after some epochs as G-27B-IT might not be a strong teacher model for Q-1.5B and it's base accuracy is already high for this task.

**Table 9:** Zero-shot and Four-shot accuracies for three models across three datasets.

| Dataset | Model | Zero-shot | Four-shot |
|---------|-------|-----------|-----------|
| MATH500 | G-2B | 0.12 | 0.17 |
| | L-8B | 0.13 | 0.19 |
| | Q-1.5B | 0.53 | 0.35 |
| GSM8K | G-2B | 0.09 | 0.25 |
| | L-8B | 0.04 | 0.39 |
| | Q-1.5B | 0.69 | 0.67 |
| Countdown | G-2B | 0.00 | 0.06 |
| | L-8B | 0.00 | 0.16 |
| | Q-1.5B | 0.00 | 0.09 |

## A.6 CoTs Analysis

---

### Example I: G-27B-IT-G

**Problem:**
A student must choose a program of four courses from a list of courses consisting of English, Algebra, Geometry, History, Art, and Latin. This program must contain English and at least one mathematics course. In how many ways can this program be chosen?

**Generated Solution by G-27B-IT leading to Correct Final Answer:**
**Step 1:** English, Algebra, and Geometry (correct)
Pick the fourth from {History, Art, Latin} — 3 ways.

**Step 2:** English, Algebra, and another math course (incorrect duplicate counts)
The student must choose one more course from the remaining three (History, Art, Latin). There are 3 ways to do this.

**Step 3:** English, Geometry, and another math course (incorrect duplicate counts)
The student must choose one more course from the remaining three (History, Art, Latin). There are 3 ways to do this.(incorrect duplicate counts)

**Step 4: Total.**
$3 + 3 + 3 = 9$
(incorrect case split with double-counting, but numerical total somehow luckily matches the correct answer by coincidence).

---

### Example I: G-27B-IT-W

**Problem:**
A student must choose a program of four courses from a list of courses consisting of English, Algebra, Geometry, History, Art, and Latin. This program must contain English and at least one mathematics course. In how many ways can this program be chosen?

**Generated Solution by G-27B-IT leading to Incorrect Final Answer:**
**Step 1:** English + Algebra + Geometry (correct)
Pick the fourth from {History, Art, Latin} — 3 ways.

**Step 2:** English + Algebra only (incorrect count)
Should exclude Geometry; pick 2 from {History, Art, Latin} $\to \binom{3}{2} = 3$. Counted as 6 instead.

**Step 3:** English + Geometry only (incorrect count)
Symmetric to Step 2; should be $\binom{3}{2} = 3$, but counted as 6.

**Step 4: Total.**
$3 + 6 + 6 = \boxed{15}$
(incorrect total; correct total should be 9).

## Example II: G-27B-IT-G

**Problem:**
Determine the value of

$$2002 + \tfrac{1}{2}\left(2001 + \tfrac{1}{2}\left(2000 + \cdots + \tfrac{1}{2}\left(3 + \tfrac{1}{2}\cdot 2\right)\right)\cdots\right).$$

**Solution (student):**
**(1) Define $S$.** (correct)

$$S = 2002 + \frac{1}{2}\left(2001 + \frac{1}{2}\left(2000 + \cdots + \frac{1}{2}\left(3 + \frac{1}{2}\cdot 2\right)\right)\cdots\right).$$

**(2) Write $\tfrac{1}{2}S$.**

$$\tfrac{1}{2}S = 1001 \;+\; \tfrac{1}{2}\left(2000 + \frac{1}{2}\left(1999 + \cdots + \frac{1}{2}\left(3 + \frac{1}{2}\cdot 2\right)\right)\cdots\right)$$

(incorrect: mis-scaled tail; halving $S$ should multiply the *entire* nested part by an extra $\tfrac{1}{2}$, not shift to start at 2000).

**(3) Subtract the two equations.**

$$\tfrac{1}{2}S = 1001 + \tfrac{1}{2}\cdot 2001 = 2001 \quad \text{(arithmetic error)}$$

**(4) Conclude $S$.**

$$S = \boxed{4002} \quad \text{(value happens to be correct, but calculation above is invalid)}$$

## Example II: G-27B-IT-W

**Problem:**
Determine the value of

$$2002 + \tfrac{1}{2}\left(2001 + \tfrac{1}{2}\left(2000 + \cdots + \tfrac{1}{2}\left(3 + \tfrac{1}{2}\cdot 2\right)\right)\cdots\right).$$

**Solution (student):**
**(1) Define $S$.** (correct)

$$S = 2002 + \frac{1}{2}\left(2001 + \frac{1}{2}\left(2000 + \cdots + \frac{1}{2}\left(3 + \frac{1}{2}\cdot 2\right)\right)\cdots\right).$$

**(2) Write $\tfrac{1}{2}S$.**

$$\tfrac{1}{2}S = 1001 \;+\; \tfrac{1}{2}\left(2000 + \frac{1}{2}\left(1999 + \cdots + \frac{1}{2}\left(3 + \frac{1}{2}\cdot 2\right)\right)\cdots\right)$$

(incorrect: the outer factor should be $\tfrac{1}{4}$ on the part beginning with 2001).

**(3) Subtract the two equations.**

$$\tfrac{1}{2}S = 1001 + \tfrac{1}{2}\cdot 2001 \;=\; 1001 + 1000.5 = 2001.5 \quad \text{(incorrect: subtraction invalid)}$$

**(4) Conclude $S$.**

$$S = \boxed{4003} \quad \text{(incorrect final value caused by the faulty subtraction)}$$

## Example III: G-27B-IT-G

**Problem:**
A student must choose a program of four courses from a list of courses consisting of English, Algebra, Geometry, History, Art, and Latin. This program must contain English and at least one mathematics course. In how many ways can this program be chosen?

**Generated Solution by G-27B-IT leading to Correct Final Answer:**
**Step 1:** English, Algebra, and Geometry (correct)
Pick the fourth from {History, Art, Latin} — 3 ways.

**Step 2:** English, Algebra, and another math course (incorrect)
Since the only other math is Geometry, this duplicates Step 1. Counted again as 3 ways.

**Step 3:** English, Geometry, and another math course (incorrect)
Since the only other math is Algebra, this duplicates Step 1. Counted again as 3 ways.

**Step 4: Total.**
$3 + 3 + 3 = 9$
(incorrect case split with double-counting, but numerical total matches the correct answer by coincidence).

## Example III: G-27B-IT-W

**Problem:**
A student must choose a program of four courses from a list of courses consisting of English, Algebra, Geometry, History, Art, and Latin. This program must contain English and at least one mathematics course. In how many ways can this program be chosen?

**Generated Solution by G-27B-IT leading to Incorrect Final Answer:**
**Step 1:** English + Algebra + Geometry (correct)
Pick the fourth from {History, Art, Latin} — 3 ways.

**Step 2:** English + Algebra only
Should exclude Geometry; pick 2 from {History, Art, Latin} $\rightarrow \binom{3}{2} = 6$.

**Step 3:** English + Geometry only $\rightarrow \binom{3}{2} = 6$

**Step 4: Total.**
$3 + 6 + 6 = \boxed{15}$
(incorrect total; correct total should be 9).

## Example I: Q-14B-IT-G

**Problem:**
Let $f(x) = \left\lceil \frac{1}{x+2} \right\rceil$ for $x > -2$, and $f(x) = \left\lfloor \frac{1}{x+2} \right\rfloor$ for $x < -2$ (undefined at $x = -2$). Which integer is *not* in the range of $f(x)$?

**Solution (model):**
**Step 1:** $x > -2 \Rightarrow \frac{1}{x+2} > 0 \Rightarrow$ range $\{1, 2, 3, \dots\}$. (correct)

**Step 2:** $x < -2 \Rightarrow \frac{1}{x+2} <= 0 \Rightarrow$ floor gives $\{0, -1, -2, -3, \dots\}$. (incorrect)

**Step 3:** No branch yields 0; thus 0 is not in the range. (incorrect last step, still comes up woth right answer)

**Final.**
$\boxed{0}$.

## Example I: Q-14B-IT-W

**Problem:**
Let $f(x) = \left\lceil \frac{1}{x+2} \right\rceil$ for $x > -2$, and $f(x) = \left\lfloor \frac{1}{x+2} \right\rfloor$ for $x < -2$ (undefined at $x = -2$). Which integer is *not* in the range of $f(x)$?

**Solution (model):**
**Step 1:** For $x > -2$, $\frac{1}{x+2} > 0$ and $\lceil \cdot \rceil$ yields $1, 2, 3, \ldots$ (correct).

**Step 2:** For $x < -2$, claim $\left\lfloor \frac{1}{x+2} \right\rfloor$ includes 0 and negatives. (incorrect)
Here $\frac{1}{x+2} < 0$, so $\lfloor \cdot \rfloor \leq -1$; 0 is not attained.

**Final.**
Missing integer $= -1$   (incorrect; correct missing integer is 0).

## Example II: Q-14B-IT-G

**Problem:**
How many integers $n$ satisfy the condition $100 < n < 200$ and the condition $n$ has the same remainder whether it is divided by 6 or by 8?

**Solution (model):**
**Step 1:** Write $n = 6k + r = 8m + r$ with $0 \leq r < 6$. (correct)

**Step 2:** Equating gives $6k + r = 8m + r \implies 3k = 4m$. (correct)

**Step 3:** Let $m = 3p \implies n = 24p + r$. (correct)

**Step 4:** Apply bounds: $100 < 24p + r < 200$. (correct)

**Step 5:** Check $r = 0, 1, 2, 3, 4$: each gives 4 values of $p$. For $r = 5$: 5 values. (correct)

**Step 6: Total.**
$4 + 4 + 4 + 4 + 4 + 5 = 25$.

**Final Answer:** $\boxed{25}$ (correct).

## Example II: Q-14B-IT-W

**Problem:**
How many integers $n$ satisfy the condition $100 < n < 200$ and the condition $n$ has the same remainder whether it is divided by 6 or by 8?

**Solution (model):**
**Step 1:** Write $n \equiv r \pmod 6$ and $n \equiv r \pmod 8$ with $0 \leq r < 6$. (correct)

**Step 2:** Then $n - r$ divisible by $\text{LCM}(6, 8) = 24 \implies n = 24k + r$. (correct)

**Step 3:** Apply bounds: $100 < 24k + r < 200$. (correct)

**Step 4:** Check each $r$: counted 4 values for each case. For $r = 5$, missed one extra value (should be 5). (incorrect)

**Step 5: Total.**
$4 + 4 + 4 + 4 + 4 + 4 = 24$.

**Final Answer:** $\boxed{24}$ (incorrect; correct total is 25).

## Countdown G-27B-IT-G — Target 57 from [50, 42, 63, 86]

**Problem:**
Using the numbers [50, 42, 63, 86], create an equation that equals 57. Use only $+, -, \times, \div$, and use each number once.

**Solution (model):**
**Step 1:** Try a difference with the largest: $86 - 50 = 36$. (correct)

**Step 2:** Combine with 63: $63 + 36 = 99$. (correct)

**Step 3:** Subtract the remaining number: $99 - 42 = 57$. (correct)

**Final.**
Expression uses all four once: $(86 - 50) + 63 - 42 = 57$. (correct)
`<answer> (86 - 50) + 63 - 42 </answer>`

## Countdown G-27B-IT-W — Target 57 from [50, 42, 63, 86]

**Problem:**
Using the numbers [50, 42, 63, 86], create an equation that equals 57. Use only $+, -, \times, \div$, and use each number once.

**Solution (model):**
**Step 1:** Consider $86 - 42 = 44$ as a starting point. (arithmetically correct)

**Step 2:** Claim "$63 - 6 = 57$" without legitimately forming 6 from the given numbers using allowed operations exactly once. (incorrect)

**Step 3:** Proposed final: $86 - 42 + 50 - 63 = 31 \neq 57$. (incorrect)

**Final.**
Does not reach 57; arithmetic and constraints violated.
`<answer> 86 - 42 + 50 - 63 </answer>` (evaluates to 31)

## GSM8K G-27B-IT-G

**Problem:**
Chelsea has 24 kilos of sugar. She divides them into 4 bags equally. Then one of the bags gets torn and half of the sugar falls to the ground. How many kilos of sugar remain?

**Solution (model):**
**Step 1:** Divide evenly: $24 \div 4 = 6$ kilos per bag. (correct)

**Step 2:** Sugar lost from torn bag: $6 \div 2 = 3$ kilos. (correct)

**Step 3:** Torn bag now has $6 - 3 = 3$ kilos left. (correct)

**Step 4:** Total remaining: $torn\ 3 + 3\ \text{bags} \times 6 = 3 + 18 = 21$. (correct)

**Final.**
Final Answer: The final answer is 21 kilos.

## GSM8K G-27B-IT-W

**Problem:**
Chelsea has 24 kilos of sugar. She divides them into 4 bags equally. Then one of the bags gets torn and half of the sugar falls to the ground. How many kilos of sugar remain?

**Solution (model):**
**Step 1:** Divide evenly: $24 \div 4 = 6$ kilos per bag. (correct)

**Step 2:** Lost from torn bag: $6 \div 2 = 3$ kilos. (correct)

**Step 3:** Torn bag left with $6 - 3 = 3$ kilos. (correct)

**Step 4:** Total remaining computed as $3 \times 3 + 3 = 12$. (incorrect)
The three *untouched* bags still have 6 kilos each, not 3; correct total is $3 + 3 \times 6 = 21$.

**Final.**
Final Answer: The final answer is 12 kilos. (incorrect)

## A.7 PROMPTS

---

### MATH PROBLEM SOLVING (ZERO-SHOT) PROMPT

**Zero-Shot Prompt Template :**

> You are a math expert. I am going to give you a math Problem that you need to solve. When you answer, respond only with the Solution, thinking step by step. At the end of the Solution, when you give your final answer, write it in the form "Final Answer: The final answer is {answer}. I hope it is correct."
>
> Problem: {problem}
> Solution:

---

### MATH PROBLEM SOLVING (FOUR-SHOT) PROMPT

You are a math expert. I am going to give you a math Problem that you need to solve. When you respond, respond only with the Solution, thinking step by step. At the end of the Solution, when you give your final answer, write it in the form "Final Answer: The final answer is {answer}. I hope it is correct."

**Problem:** Find the domain of the expression $\frac{\sqrt{x-2}}{\sqrt{5-x}}$.
**Solution:** The expressions inside each square root must be non-negative. Therefore, $x - 2 \geq 0$, so $x \geq 2$, and $5 - x \geq 0$, so $x \leq 5$. Also, the denominator cannot be equal to zero, so $5 - x > 0$, which gives $x < 5$. Therefore, the domain of the expression is $\boxed{[2, 5)}$. Final Answer: The final answer is $[2, 5)$. I hope it is correct.

**Problem:** If $\det \mathbf{A} = 2$ and $\det \mathbf{B} = 12$, then find $\det(\mathbf{AB})$.
**Solution:** We have that $\det(\mathbf{AB}) = (\det \mathbf{A})(\det \mathbf{B}) = (2)(12) = \boxed{24}$. Final Answer: The final answer is $24$. I hope it is correct.

**Problem:** Terrell usually lifts two 20-pound weights 12 times. If he uses two 15-pound weights instead, how many times must Terrell lift them in order to lift the same total weight?
**Solution:** If Terrell lifts two 20-pound weights 12 times, he lifts a total of $2 \cdot 12 \cdot 20 = 480$ pounds of weight. If he lifts two 15-pound weights instead for $n$ times, he will lift a total of $2 \cdot 15 \cdot n = 30n$ pounds of weight. Equating this to 480 pounds, we can solve for $n$:

$$30n = 480$$

$$\Rightarrow \quad n = 480/30 = \boxed{16}$$

Final Answer: The final answer is $16$. I hope it is correct.

**Problem:** If the system of equations

$$6x - 4y = a,$$
$$6y - 9x = b.$$

has a solution $(x, y)$ where $x$ and $y$ are both nonzero, find $\frac{a}{b}$, assuming $b$ is nonzero.
**Solution:** If we multiply the first equation by $-\frac{3}{2}$, we obtain $6y - 9x = -\frac{3}{2}a$. Since we also know that $6y - 9x = b$, we have $-\frac{3}{2}a = b \Rightarrow \frac{a}{b} = \boxed{-\frac{2}{3}}$. Final Answer: The final answer is $-\frac{2}{3}$. I hope it is correct.

**Problem:**
{problem}

**Solution:**

## GSM8K PROBLEM SOLVING (ZERO-SHOT) PROMPT

You are a math expert. I am going to give you a math Problem. Think step by step and you generate the solution. Write the final answer in the form "Final Answer: The final answer is #### answer."

**Problem:**
{problem}

**Solution:**

## GSM8K PROBLEM SOLVING (FOUR-SHOT) PROMPT

You are a math expert. I am going to give you a math Problem. Think step by step and you generate the solution. Write the in the final answer in the form "Final Answer: The final answer is #### answer."

**Problem:** Janet's ducks lay 16 eggs per day. She eats three for breakfast every morning and bakes muffins for her friends every day with four. She sells the remainder at the farmers' market daily for $2 per fresh duck egg. How much in dollars does she make every day at the farmers' market?
**Solution:** Janet sells 16 - 3 - 4 = «16-3-4=9»9 duck eggs a day.
She makes 9 * 2 = $«9*2=18»18 every day at the farmer's market.
**Final Answer:** The final answer is #### 18.

**Problem:** A robe takes 2 bolts of blue fiber and half that much white fiber. How many bolts in total does it take?
**Solution:** It takes 2/2=«2/2=1»1 bolt of white fiber.
So the total amount of fabric is 2+1=«2+1=3»3 bolts of fabric.
**Final Answer:** The final answer is #### 3.

**Problem:** Josh decides to try flipping a house. He buys a house for $80,000 and then puts in $50,000 in repairs. This increased the value of the house by 150%. How much profit did he make?
**Solution:** The cost of the house and repairs came out to 80,000+50,000=$«80000+50000=130000»130,000.
He increased the value of the house by 80,000*1.5=«80000*1.5=120000»120,000.
So the new value of the house is 120,000+80,000=$«120000+80000=200000»200,000.
So he made a profit of 200,000-130,000=$«200000-130000=70000»70,000.
**Final Answer:** The final answer is #### 70000.

**Problem:** James decides to run 3 sprints 3 times a week. He runs 60 meters each sprint. How many total meters does he run a week?
**Solution:** He sprints 3*3=«3*3=9»9 times.
So he runs 9*60=«9*60=540»540 meters.
**Final Answer:** The final answer is #### 540.

**Problem:**
{problem}

**Solution:**

## COUNTDOWN PROBLEM SOLVING (ZERO-SHOT) PROMPT

You are a math expert. I am going to give you a Problem that you need to solve. When you respond, respond with the Solution, thinking step by step. And return the final answer in `<answer> </answer>` tags, for example `<answer> (1 + 2) / 3 </answer>`.

**Problem:**
Using the numbers {nums}, create an equation that equals {target}. You can use basic arithmetic operations (+, -, *, /) and each number can only be used once. Do not use any other operations or numbers.

**Solution:**

## COUNTDOWN PROBLEM SOLVING (FOUR-SHOT) PROMPT

You are a math expert. I am going to give you a Problem that you need to solve. When you respond, respond with the Solution, thinking step by step. And return the final answer in `<answer> </answer>` tags, for example `<answer> (1 + 2) / 3 </answer>`.

**Problem:** Using the numbers [38, 98, 56, 14], create an equation that equals 91. You can use basic arithmetic operations (+, -, *, /) and each number can only be used once. Do not use any other operations or numbers.
**Solution:** I am looking for a combination of numbers and operations that results in 91. I can try to combine numbers using addition first. Let's try adding 38 and 14. 38 + 14 = 52. Now I have the numbers 52, 98, and 56 left to use. I need to get to 91. Let's see if multiplication or division can help. Let's try multiplying 52 by 98. 52 * 98 is a large number. Let's try dividing by 56. So, (52 * 98) / 56. I can simplify this calculation. 98 and 56 are both divisible by 14. 98 / 14 = 7. 56 / 14 = 4. So, the expression becomes 52 * (7 / 4). I can rewrite this as (52 / 4) * 7. 52 / 4 = 13. Now I just need to multiply 13 by 7. 13 * 7 = 91. This gives the target number. The full equation is ((38 + 14) * 98) / 56. `<answer> ((38 + 14) * 98) / 56 </answer>`

**Problem:** Using the numbers [23, 63, 79, 51], create an equation that equals 68. You can use basic arithmetic operations (+, -, *, /) and each number can only be used once. Do not use any other operations or numbers.
**Solution:** I am looking for a combination of numbers and operations that results in 68. I'll start with the largest number, 79. To get to 68, I need to subtract 11. Can I make 11 from 23, 63, and 51? 63 - 51 = 12. This is close to 11. Let's try 79 - (63 - 51) = 79 - 12 = 67. This is very close to 68, but not exactly. Let's try another combination. 79 + 51 = 130. 63 + 23 = 86. 130 - 86 = 44. Let's try another path. 63 - 23 = 40. 79 - 51 = 28. 40 + 28 = 68. This works! I have found a solution. The steps are: subtract 23 from 63 to get 40. Subtract 51 from 79 to get 28. Add the results together. `<answer> (63 - 23) + (79 - 51) </answer>`

**Problem:** Using the numbers [16, 17, 58], create an equation that equals 91. You can use basic arithmetic operations (+, -, *, /) and each number can only be used once. Do not use any other operations or numbers.
**Solution:** I need to reach the target of 91 using the numbers 16, 17, and 58. Since there are only three numbers, I'll try adding them up first. 58 + 17 = 75. Now, I need to incorporate the last number, 16. 75 + 16 = 91. This is the target number. So, the solution is to add all the numbers together. `<answer> 58 + 17 + 16 </answer>`

**Problem:** Using the numbers [2, 28, 78], create an equation that equals 11. You can use basic arithmetic operations (+, -, *, /) and each number can only be used once. Do not use any other operations or numbers.
**Solution:** The target is 11. The numbers are 2, 28, and 78. The numbers are quite spread out, so simple addition or subtraction of all of them at once is unlikely to work. Let's see if there is a division or multiplication that simplifies the problem. 78 is an even number, so it's divisible by 2. 78 / 2 = 39. Now I have the number 39, and the remaining number is 28. I need to reach the target of 11. Let's see the difference between 39 and 28. 39 - 28 = 11. This is the target number. So the equation is (78 / 2) - 28. `<answer> (78 / 2) - 28 </answer>`

**Problem:**
Using the numbers {nums}, create an equation that equals {target}. You can use basic arithmetic operations (+, -, *, /) and each number can only be used once. Do not use any other operations or numbers.

**Solution:**

---

### Prompt for Math Paraphrasing

You are a math expert. I am going to give you a Problem and a correct solution (provided as a hint). Your task is to rewrite the solution in your own words and style, while making use of the hint as guidance.
Ensure that your reasoning follows the same logical steps as the hint, and that your final answer matches the solution's final result. You do not need to generate a new solution, just rewrite in your own style while adhering to the hint solution.

**Problem:**
problem

**Correct Solution (provided as hint) to be Paraphrased:**
response

**Your Paraphrased Solution:**

---

### PROMPT FOR GENERATING COMPLETELY FLAWED REASONING

You have been given a math problem. Your task is to create a completely flawed mathematical solution to the problem below that results in an incorrect answer. Every step must use invented formulas and incorrect reasoning. However, ensure that the solution should not be completely random and the solution should be based on the given problem. Do NOT mention that you are generating an incorrect solution anywhere in the solution as this data will be used for error analysis research.
Final Answer Format: At the end of the Solution, when you give your final answer, write it in the form Final Answer: The final answer is $answer$. I hope it is correct.

**Problem**: {problem}

**Completely Incorrect Solution**:

---

## A.8 STATISTICAL SIGNIFICANCE AND HYPERPARAMETER ABLATIONS

To ensure the robustness of our results, we performed experiments on the Gemma 2B model on the Human H CoTs and Gemma-27B-IT generated G and W CoTs, and we used two complementary measures: (i) we report mean $\pm$ standard deviation across 5 random seeds to characterize run-to-run variability, and (ii) we report 95% confidence intervals on the mean (t-based over 5 seeds) to quantify the uncertainty of the estimated average performance. Table 10 shows that our results are robust across the runs. We report results on the max-scores that we report in our tables. But the trends hold across all checkpoints across all 5 runs.

**Table 10:** Summary of robustness metrics over 5 random seeds. We report mean $\pm$ standard deviation and 95% confidence intervals on the mean. We run the experiments on Gemma 2B (G2B) over the datasets- H, G27B-IT-G, and G27B-IT-W.

| Run | Mean $\pm$ Std Dev | 95% CI on Mean |
|---|---|---|
| G2B on H | $0.162 \pm 0.005$ | $0.162 \pm 0.006$ |
| G2B on G27B-IT-W | $0.183 \pm 0.008$ | $0.183 \pm 0.009$ |
| G2B on G27B-IT-G | $0.223 \pm 0.005$ | $0.223 \pm 0.006$ |

We also report a detailed ablation varying the batch size and the learning rate to study the results when the two main hyperparameters of our experiments change. We again performed experiments on Gemma 2B (G2B) with Human H CoTs and Gemma-27B-IT generated G and W CoTs. We ablated the batch size over 16, 64, and 256, and the learning rate over 2e-5 and 1e-6. We present the results in Table 11 below.

**Table 11: Ablation of result trends across different hyperparameters.** We perform a detailed ablation of the main hyperparameters- batch size (BS) and learning rate (lr) over the Gemma-2B model on the MATH related datasets to show that the trends of results that synthetic G and W CoTs outperform H CoTs remain consistent across all runs. We provide maximum accuracy on MATH500 test set for G-2B for all these runs.

| Model | H | G-27B-IT-W | G-27B-IT-G |
|---|---|---|---|
| BS 256 lr 2e-5 | 0.17 | +0.02 | +0.04 |
| BS 256 lr 1e-6 | 0.14 | +0.03 | +0.05 |
| BS 64 lr 2e-5 | 0.17 | +0.02 | +0.06 |
| BS 64 lr 1e-6 | 0.15 | +0.02 | +0.04 |
| BS 16 lr 2e-5 | 0.15 | +0.02 | +0.05 |
| BS 16 lr 1e-6 | 0.17 | +0.02 | +0.02 |

## A.9 RESULTS AND ANALYSIS ON NON-MATHEMATICAL REASONING TASK - CODE GENERATION

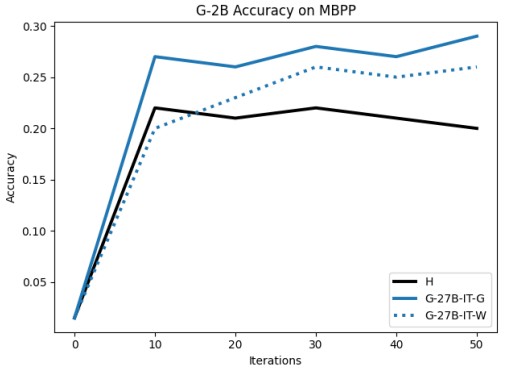

**(a)** Performance of G-2B on MBPP test set using our different datasets for finetuning (64 batch size, lr=2e-5)

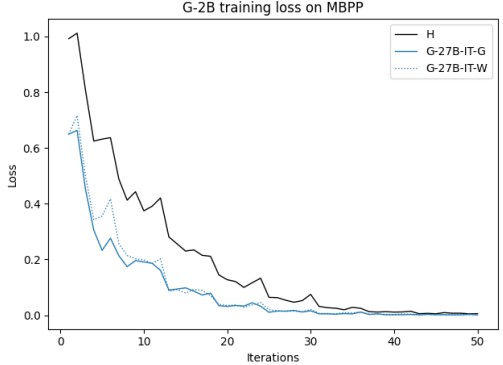

**(b)** Training losses on the corresponding datasets hold similar trend with H being distributionally farther away from the base model than G and W.

**Figure 15:** G and W outperforms H. Training losses clearly show how H being further away from the model's distribution as compared to the synthetic datasets lead to similar results even in other reasoning domains like code generation generalizing beyond math reasoning tasks.

To test the generalizability and strength of our method of learning from synthetic CoTs including those leading to incorrect answers, beyond mathematical reasoning tasks, we experimented with code generation. We used the MBPP dataset (Austin et al., 2021) and kept aside a subset of 200 programming tasks as the test set on which we evaluate Pass@1 scores using temperature t=0 (greedy decoding). We used remaining 774 programming problems and following our standard method of synthetic data generation as in other tasks using G-27B-IT, generated a dataset of programming solutions leading to correct solutions passing all test cases (G-27B-IT-G) and compilable incorrect solutions failing one or more test cases (G-27B-IT-W). We provide the prompt used for program generation below. We ended up taking a subset of 354 datapoints for H, G and W. This is due the fact that many samples had all the incorrect and all the correct final answer CoTs, making the overlapping datasets size small. We perform the experiment on G-2B model. As we can see from the results in Table 12, both W and G achieve max accuracies higher than the H baseline by 4% and 7% respectively. Figure 15b again shows how this results is also complemented by the fact that H is distributionally farther away indicated by the clearly higher training loss consistently over the SFT than W and G. Additionally, we conduct the paraphrasing experiment where we paraphrase H solutions using G-27B-IT model. As shown in Figure 16a, the performance improves with 6% increase in the absolute accuracy and the loss curves (shown in Figure 16b) show that the paraphrased CoTs are closer to the distribution of the model. This experiment strongly supports our finding: CoTs leading to correct or incorrect final answers can outperform entirely correct human written

CoTs if they are closer to the model's distribution. We also provide the prompt for this code solution paraphrasing experiment below.

**Table 12: Max accuracy on MBPP**. Both **W** and **G** CoT programming traces from stronger model (G-27B-IT) outperform **H** CoT traces after SFT on corresponding datasets. Moreover Paraphrased H dataset performs much better highlighting the importance of distribution of dataset.

| Model | H | G-27B-IT-W | G-27B-IT-G | H-Para-G-27B-IT |
|-------|------|------------|------------|-----------------|
| G-2B | 0.22 | +0.04 | **+0.07** | +0.06 |

---

### MBPP CODE GENERATION PROMPT

You are an expert Python programmer. I will give you a programming task description. Your job is to write a correct, efficient, and clean Python solution. Start directly with the coding solution.

Requirements: - Use only the Python standard library. - Your code must strictly satisfy the provided assertion. - Respond with ONLY Python code (no backticks, no comments, no explanations).

**Problem:**
{problem}

Your code should satisfy the following assertion:
{assertion}

**Solution:**

---

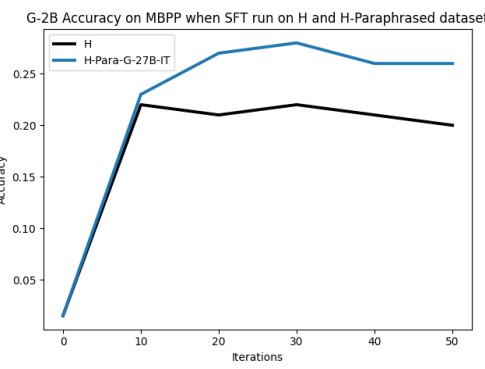

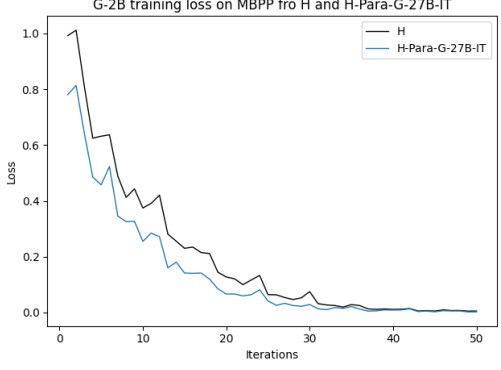

**(a)** Performance of G-2B on MBPP test set using H and Paraphrased version of H using G-27B-IT (64 batch size, lr=2e-5)

**(b)** Training losses on the corresponding datasets hold similar trend with H being distributionally farther away from the base model than its paraphrased version H-Para-G-27B-IT.

**Figure 16:** G and W outperforms H. Training losses clearly show how H being further away from the model's distribution as compared to the synthetic datasets lead to similar results even in other reasoning domains like code generation generalizing beyond math reasoning tasks.

## Prompt for Code Paraphrasing

You are an expert Python programmer. I am going to give you a full programming task description (including assertions) and a working correct solution (provided as a hint). Your task is to rewrite this correct code solution in your own style while **STRICTLY** preserving the exact logic, functionality, and function signature of the provided hint solution. You do not need to generate a new solution, just rewrite in your own coding style while adhering to the hint solution.

**Problem:**
problem

**Correct Solution (provided as hint) to be Paraphrased:**
response

**Your Paraphrased Solution:**

