# OpenReview forum: "Shape of Thought: When Distribution Can Matter More than Correctness in Reasoning Tasks"
_ICLR.cc/2026/Conference — ICLR 2026 Conference Desk Rejected Submission_

### Official Review · Reviewer_BDoS · 2025-10-27

**Soundness:** 2
**Presentation:** 3
**Contribution:** 2
**Rating:** 4
**Confidence:** 4

**Summary:**

This work explores how distillation from the reasoning of larger models can provide useful training signal to smaller models even when the distillation data is noisy (the reasoning from the larger model yields wrong answers).  Surprisingly, this noisy data can train the model to perform better on the task than human-curated data (human-annotated "chains-of-thought").  This is shown by having Gemma-2-27B-IT generate traces for reasoning benchmarks such as GSM8k, MATH500, and Countdown. These traces are separated into two datasets, one where the model got the answer correct, "G", and one where the model got the answer incorrect, "W". The authors then do SFT on these traces, including an additional dataset of human-curated data, "H".  The authors then evaluate these tasks and show that datasets with incorrect traces sampled from stronger models often outperform human-curated datasets. Additionally the authors show that by paraphrasing the human-curated data by the larger model, most of this performance gain can be explained, suggesting that much of the weakness from human curated data is from it being so far out of distribution from the model being trained.

The authors additionally show that as traces becoming more incorrect (i.e. corrupting traces with models to be increasingly more incorrect), the performance gains slowly degrade. This is paired with a small qualitative analysis suggesting that the reason incorrect traces provide strong signal is because each trace may contain individual steps that are correct and thus provide signal to the weaker model.

**Strengths:**

- I like the hypothesis and the comparison to human-curated CoTs is fascinating! The results for GSM8k and Math500 are also pretty compelling (the main figures).
- Good set of models are being tested and evaluated.

**Weaknesses:**

1. **The tables and figures feel either cherry-picked or are missing important entries (see below)**
- Figure 2, why is there no human baseline for the Qwen2.5-1.5B model?
- Why is there no human dataset for countdown? Without it, the paragraph "Countdown as an Out-of-distribution Task." means little to me -- I don't think this paper is arguing that incorrect traces teach the model just as much as using correct traces. So of course correct traces will yield improvements over "W".
- Figure 5 in the appendix shows more results but I think "H" is still it's own line when, if I understood correctly, "H" should be 1 line per model (you cannot compare "H" trained on G-2B with the Qwen model because that's not a fair comparison).
- (minor) please put in the actual performances and accuracies, not deltas. (i.e. table 2 and 3)

2. **Most of these findings have been found in "LLMs Can Easily Learn to Reason from Demonstrations
Structure, not content, is what matters!" and I would encourage the authors to cite this work**. It's fairly well known and does a far more indepth analysis on the corruption of long chains-of-thought.  A significant difference to this papers credit is the inclusion of human data and that this paper seems to be looking at "CoT" pre-R1 (so not "long cots" as the paper I mentioned above explores).

3. **The main conclusions drawn are weakly supported.**
- a. ": (i) synthetic traces are closer to the student model’s output distribution and are thus easier to learn from", there was not an experiment that directly tested this. Table 4 is showing how "G", "W", and "H" all effect models at scale (looking only at G-2B vs G-9B for Math500, so this may not be supported on the other models or tasks). And Table 5 shows lower starting training loss, but again is only on Math500 and G-2B.  *The core finding of this paper is that human curated data is often weaker than noisy model generated data*, these other findings need controlled experiments with multiple tasks and models to be established.
- b. " (ii) “wrong” traces frequently contain reusable, partially correct reasoning steps." - this is supported by the findings in section 5.5 but the authors only look at a total of 40 traces to draw this conclusion. I don't think this was substantially found. The notion of "completely incorrect" in section 5.4 is also strenuous at best, "We prompt the G-27B-IT model to generate completely flawed CoTs using
MATH training problems." -- what does this mean? A "completely flawed CoT" could mean utter gibberish, or it invented a whole new problem to solve, or it could be exactly the same as those that were in "W". So what should we gather from this finding? That your distillation data doesn't have to be correct but does need to be incorrect in some specific way to be useful? I think more analysis and more controlled experiments are needed, check "LLMs Can Easily Learn to Reason from Demonstrations Structure, not content, is what matters!" for examples of the type of controlled errors I am thinking of. And to be clear, I saw the appendix section too. While explaining the current setup would be helpful, I think this whole experiment needs more control on what the "errors" are, otherwise I cannot detangle these from the previous experiments ("W" datasets, etc.) prompting for completely inaccurate traces seems far to open ended to draw any useful conclusions.



**minor notes**: significance testing would be nice to see here because the deltas can be so small, please use "Figure 2" or "Appendix H" etc. not just "2" or "H" etc. Why was 4-shot ICL included in the evaluations? It felt completely out of place. Please show all results for all models and datasets, it may be a large table, but would make a lot of what is being said clearer.

**Questions:**

See my weaknesses for questions per point.

---

> ### Author Response · Authors · 2025-11-21
> **Author Response (1/3)**
>
> **[1] The tables and figures feel either cherry-picked or are missing important entries.**
>
> We would like to clarify that the results and figures we show are not cherry-picked. We address the reviewer's concerns by answering the related questions in detail.
>
>
> **[1.1] Figure 2, why is there no human baseline for the Qwen2.5-1.5B model?**
>
> As there is no known human written CoT dataset for Countdown, we do not provide a human baseline for Countdown tasks. We study Countdown in an entirely synthetic data regime.
>
> **[1.2] Why is there no human dataset for countdown?**
>
> We could not find human written CoTs for countdown. However, if the reviewer knows a reference where we can get the same, we are happy to include the same in our experiments.
>
> **[1.3] Without it, the paragraph "Countdown as an Out-of-distribution Task." means little to me -- I don't think this paper is arguing that incorrect traces teach the model just as much as using correct traces. So of course correct traces will yield improvements over "W".**
>
> We agree with the reviewer that saying “Countdown is an Out-of-distribution task” is a strong claim. We meant to convey that Countdown is a harder task as compared to the other reasoning tasks we experiment with, given 0% zero-shot and very low few-shot accuracy of the base models as shown in table 9. We have addressed the same in the revised pdf where we mention Countdown as a “harder” task. We hypothesize that for highly procedural and constrained tasks like Countdown, learning  signals from 'W' traces are weaker than “G” as compared to more open-ended math problems where this gap seems low. We discuss the same in Section 5.1.
>
>
> **[1.4] Figure 5 in the appendix shows more results but I think "H" is still it's own line when, if I understood correctly, "H" should be 1 line per model (you cannot compare "H" trained on G-2B with the Qwen model because that's not a fair comparison).**
>
> This is a typo, we apologize for the same. In Figure 5 (a), Q-1.5B-G should be corrected to G-2B-G and Q-1.5B-W should be corrected to G-2B-W. These are G-2B fine tuning results on the synthetic datasets generated using the same base model. The datasets were generated using the 4-shot prompting using the G-2B model. We have corrected the same in the revised version. We are happy to clarify if the reviewer has more questions.
>
>
> **[1.5] please put in the actual performances and accuracies, not deltas.**
>
> We’ll address the suggestion in the final version of the paper.
>
>
> **[2] Most of these findings have been found in "LLMs Can Easily Learn to Reason from Demonstrations Structure, not content, is what matters!" and I would encourage the authors to cite this work. ...**
>
> We thank the reviewer for highlighting the related work (Li et al., 2025). We have cited this paper in the revised manuscript (refer Section 3). Below, we clarify the key differences:
>
> *Distribution vs Structure*: Li et al. argue that the structure of long CoTs is a critical factor for robust reasoning performance. They describe structure as “logical consistency across steps”. However, we investigate how the distributional alignment of short CoTs to the model’s own output distribution impacts performance on both reasoning and algorithmic benchmarks, even when CoTs have similar structure. To clarify this distinction, we randomly sampled 10 CoTs from each of the H, G and W datasets (these CoTs are uploaded with the supplementary material) and inspected their stepwise structure—all show similar logical formats. However, when we compute perplexity under the model being finetuned, the distributions differ substantially.
>
>
> ### Quantitative Perplexity Analysis of 10-CoTs
>
> Quantitative Perplexity analysis of the 10-CoTs with identical structure that we qualitatively analyzed.
>
> | Data | Perplexity |
> | :--- | :--- |
> | H | 4.092 |
> | G27B-IT-G | 2.906 |
> | G27B-IT-W | 2.968 |
>
>
> *Short vs Long CoTs*: Li et al. study long CoTs containing behaviors such as verification, backtracking, and reflection. As the reviewer notes, our work focuses on short, pre-R1 CoTs, which lack these behaviours and consists of only logical steps. Because long CoTs can be viewed as extending these short CoTs with auxiliary reflective behaviors, we expect our findings on short-CoT distributional alignment to carry over to the long-CoT regime as well.
>
> *Comparison with Human CoTs*: Li et al. evaluate only synthetic traces. In contrast, as the reviewer already mentions, a core contribution of our work is a direct comparison against fully correct, human-curated CoTs, which are widely considered “gold-standard”. Our experiments show that without distributional alignment, even carefully curated human data can underperform imperfect synthetic traces aligned to the model’s distribution. Furthermore, with paraphrasing experiments, we show that paraphrased data beats original human data, supporting our distributional alignment hypothesis.

---

> ### Author Response · Authors · 2025-11-21
> **Author Response (2/3)**
>
> **[2] continued...**
>
> *Algorithmic Tasks Reveal Limits of Structure-Only Explanations*: On algorithmic tasks such as Countdown, we observe substantial degradation when finetuning on W traces compared to G traces. As mentioned in Section 5.1, this task requires a logically precise sequence of operations to reach the exact target number. Li et al. do not study this boundary condition. Therefore, their structural hypothesis can be viewed as a special case of our broader distributional alignment framework.
>
> *Error Tolerance Experiments*: While Li et al. explore corruption of long CoTs, their perturbations (digit changes, step shuffling, etc.) might push the corrupted trace away from the model’s output distribution. Our corruption procedures are intentionally designed to preserve distributional similarity, allowing us to isolate how much distribution-aligned noise a model can tolerate.
>
> **[3] The main conclusions drawn are weakly supported.**
>
> We provide additional evidence to support the reviewer's concerns.
>
> **"a. (i) synthetic traces are closer to the student model’s output distribution and are thus easier to learn from", there was not an experiment that directly tested this. Table 4 is showing how "G", "W", and "H" all effect ... these other findings need controlled experiments with multiple tasks and models to be established.**
>
> *Loss curves*: We provide the loss curves for the GSM8K experiments across the models and the datasets in the revised pdf. Please refer to Figure 14 in the Appendix for the loss curves for all the training iterations. Loss curves corresponding to Table 5 are shown in Figure 10 (b) in Appendix.
>
> Loss trends clearly show high losses corresponding to H data as compared to W or G data, showing that H data is away from the model’s distribution as compared to W or G data. Since W/G data are closer to the model's distribution, they are more conducive to learning even if they are not fully correct (refer Tables 2 and 3) , supporting our claims.
>
> *Paraphrasing experiment*: In addition to the paraphrasing experiment with Gemma models (refer Table 4) where we finetuned Llama-3.1-8B model with human data paraphrased by Llama-3.1-70B-Instruct model. We observe substantial gains (absolute increase of 7% accuracy) as shown in the table below. Refer to Figure 13 in the Appendix for the performance across all the iterations. These high gains strongly suggest that bringing the CoTs closer to the model’s distribution can significantly improve the performance, directly addressing the reviewer’s concerns.
>
> | Model | **H** | **H-Para-L70B-IT** |
> | :--- | :--- | :--- |
> | **L-8B** | 0.19 | 0.26 (+0.07) |
>
>
> **b. " (ii) “wrong” traces frequently contain reusable, partially correct reasoning steps." - this is supported by the findings in section 5.5 but the authors only look at a total of 40 traces to draw this conclusion. ... prompting for completely inaccurate traces seems far to open ended to draw any useful conclusions.**
>
> *Qualitative Analysis of CoTs*: We agree that analyzing 40 CoTs is a small-scale qualitative study, and we acknowledge this limitation in the paper. However, we emphasize that these CoTs were randomly sampled from the dataset and not hand-selected. Performing a large-scale human evaluation was not feasible due to resource constraints, and we therefore release all datasets to facilitate future large-scale analysis. We also note that automatically verifying CoT correctness is itself an open research challenge, which we discuss in our Future Works and Limitations section.
>
>
> *Completely Flawed CoTs*: To clarify what we mean by “completely flawed” CoTs, we include the exact prompt used for their generation in the paper (Appendix, p. 28). The prompt explicitly instructs the model to invent incorrect formulas and reasoning steps while still producing a coherent, problem-specific solution rather than random text. For convenience, we reproduce the prompt below.

---

> ### Author Response · Authors · 2025-11-21
> **Author Response (3/3)**
>
> *Prompt*: You have been given a math problem. Your task is to create a completely flawed mathematical
> solution to the problem below that results in an incorrect answer. Every step must use invented
> formulas and incorrect reasoning. However, ensure that the solution should not be completely
> random and the solution should be based on the given problem. Do NOT mention that you
> are generating an incorrect solution anywhere in the solution as this data will be used for error
> analysis research.
> Final Answer Format: At the end of the Solution, when you give your final answer, write it in
> the form Final Answer: The final answer is $answer$. I hope it is correct.
>
> *Example output of the flawed CoTs.*
>
> Problem:
> 24 lemons are required to make 32 gallons of lemonade. How many lemons are needed to make 4 gallons
> of lemonade?
> Solution (model):
> First, we need to find the lemon-to-gallon ratio. We can do this by dividing the number of lemons by the
> number of gallons, then multiplying by the square root of the number of lemons. Lemon-to-gallon ratio =
> (24 lemons / 32 gallons) * √24 = 0.75 * 4.899 = 3.674 Now, to find the number of lemons needed for 4
> gallons, we multiply the lemon-to-gallon ratio by the desired number of gallons and then divide by the
> cube root of the desired number of gallons. Lemons needed = 3.674 * 4 gallons / 3
> √4 = 14.696 / 1.587 =
> 9.26 Final Answer: The final answer is 9.26. I hope it is correct.
>
> We also show an example flawed CoT in the Appendix (p. 13). As illustrated in the example, the model generates structured, readable reasoning that stays on-topic but uses mathematically invalid transformations (e.g., multiplying by √24 or dividing by the cube root of the target gallons). These traces are thus qualitatively different from the W dataset:
> - W: model attempts to solve the problem but arrives at an incorrect answer.
> - Flawed CoTs: systematically constructed incorrect reasoning using violated rules and invented formulas, while remaining coherent and problem-aligned.
>
> We release the entire flawed-CoT dataset in the supplementary material for transparency and future evaluation.
>
> *Why flawed CoTs are relevant to the error-tolerance experiments*:
>  In our error-tolerance setup, we intentionally introduce errors that preserve the distributional properties of the CoTs as measured under the student model. As shown in Figure 10(b) (Appendix), flawed CoTs exhibit loss values similar to W and even G, indicating that they remain close to the model’s output distribution despite being mathematically incorrect. This is essential for the experiment: the goal is to study how much distribution-aligned noise a model can tolerate while still outperforming human-curated CoTs.
>
> Importantly, Li et al.’s corruption procedures—although valuable—introduce synthetic errors (digit swaps, step shuffling) that can shift the CoT outside the model’s natural distribution. Such perturbations would confound our study, whose purpose is to isolate the role of distributional alignment rather than explore structural corruption in general.
>
> *Main takeaway of the experiment*:
>  Even when replacing 25% of G CoTs with flawed CoTs, the model still outperforms training on human CoTs alone (Table 5; Section 5.4). This demonstrates that:
> - correctness is not the primary driver of learnability,
> - distributional alignment can outweigh perfect correctness, and
> - models can tolerate surprisingly large amounts of structured, distribution-aligned noise.
>
> **significance testing would be nice to see here because the deltas can be so small, please use "Figure 2" or "Appendix H" etc. not just "2" or "H" etc. Why was 4-shot ICL included in the evaluations? It felt completely out of place. Please show all results for all models and datasets, it may be a large table, but would make a lot of what is being said clearer.**
>
> Please refer to the common response for the significant testing. Table 6 presents the major results. We are happy to conduct more experiments if the reviewer suggests the same.
>
> -----------------------------------------------------
> We would like to gently remind the reviewer to let us know if there are any further questions or feedback.
>
> We also respectfully request that the reviewer reconsider the score in light of the clarifications we have provided. We remain happy to engage in further discussion to address any additional concerns or questions.

---

> > ### Author Response · Authors · 2025-12-02
> > **Author Response**
> >
> > **Update on [2]**
> >
> > We have added new experiments on code generation which strongly support our findings and makes them distinct as detailed in the common response. We confirm that these new results (currently in Appendix A.9) will be moved to the main paper in the final version.

---

### Official Review · Reviewer_Vexn · 2025-10-28

**Soundness:** 3
**Presentation:** 3
**Contribution:** 3
**Rating:** 4
**Confidence:** 4

**Summary:**

The authors show that, across MATH, GSM8K and Countdown, supervised fine-tuning (SFT) on chain-of-thought traces that reach a wrong final answer can outperform training on correct human-written traces. Experiments are extensive (3 tasks × 4 model sizes × 3 data types) and reproducibility is good (code, prompts, checkpoints promised).

**Strengths:**

At multiple model families & scales, trends hold across all, indicating generality within the math domain.

The paper offers a careful ablation of distribution vs. correctness.

**Weaknesses:**

The core finding of this paper is no longer novel.

All experiments on math word problems; no evidence for code, commonsense, or scientific reasoning.

No step-level F1 or human-verified correctness; claims about “faithfulness” rest on 20 hand-checked examples.

Hyper-parameter sensitivity and statistical significance are not explored.

Only one 9B model; no 30B+ or frontier-scale teacher.

No comparison with RL or verifier-based filtering

**Questions:**

Run identical pipeline on CodeForces (code) and ARC-c (science); report macro-averaged gain.

Implement R1-style RL with outcome reward on same compute; compare final accuracy and wall-clock cost.

---

> ### Author Response · Authors · 2025-11-21
> **Author Response (1/1)**
>
> **[1] The core finding of this paper is no longer novel.**
>
> We respectfully note that the core contributions of our work remain novel, as detailed in our common response outlining the technical contributions and the precise aspects in which our approach differs from prior work. If the reviewer is aware of specific papers that directly address the same research question, we would be grateful if they could point them out.
>
> **[2] All experiments on math word problems; no evidence for code, commonsense, or scientific reasoning.**
>
> While our main evaluations focus on math word problems, we also include experiments on algorithmic reasoning tasks such as Countdown. As stated in our limitations section, we explicitly acknowledge that transfer to domains such as code and scientific reasoning is an important direction for future work.
>
> **[3] No step-level F1 or human-verified correctness; claims about “faithfulness” rest on 20 hand-checked examples.**
>
> We qualitatively show that with the red colored steps being incorrect and green colored steps being correct for some CoTs we analysed (refer A.6 CoTs Analysis). Due to time constraints, we couldn’t show the same for all the CoTs.
>
> **[4] Hyper-parameter sensitivity and statistical significance are not explored.**
>
> Please refer to the common response for the same, we have added results for both.
>
> **[5] Only one 9B model; no 30B+ or frontier-scale teacher.**
>
> Due to compute constraints, we cannot run for larger than 9B models. Frontier-scale teacher CoTs have stylistical differences from those of short CoTs we study and might confound our study.
>
> **[6] No comparison with RL or verifier-based filtering**
>
> RL is beyond the scope of our study, as we study what datasets are conducive for supervised finetuning. Our aim was not to maximize reasoning performance using RL finetuning or verifier-guided filtering for SFT, which are not related to our current scope.
>
> **[7] Run identical pipeline on CodeForces (code) and ARC-c (science); report macro-averaged gain.**
>
> Due to time and compute constraints, we cannot run the experiments on more domains. We believe that we demonstrate our findings with a good number of tasks (as pointed out by other reviewers). We do mention studying more tasks in future works.
>
> **[8] Implement R1-style RL with outcome reward on same compute; compare final accuracy and wall-clock cost.**
>
> This is clearly beyond the scope of our work.
>
> -----------------------------------------------------
> We would like to gently remind the reviewer to let us know if there are any further questions or feedback.
>
> We also respectfully request that the reviewer reconsider the score in light of the clarifications we have provided. We remain happy to engage in further discussion to address any additional concerns or questions.

---

> > ### Author Response · Authors · 2025-12-02
> > **Author Response**
> >
> > **Update on [2]**
> >
> > We have added new experiments on code generation which strongly support our findings, as detailed in the common response. We confirm that these new results (currently in Appendix A.9) will be moved to the main paper in the final version.

---

### Official Review · Reviewer_KHFM · 2025-10-30

**Soundness:** 2
**Presentation:** 2
**Contribution:** 2
**Rating:** 4
**Confidence:** 4

**Summary:**

This paper investigates a hypothesis that training language models on incorrect chain-of-thought (CoT) reasoning traces can still improve reasoning performance, sometimes even outperforming training on human-annotated CoTs. The authors conduct extensive experiments across three reasoning benchmarks (MATH, GSM8K, Countdown) and multiple model families (Gemma, Llama, Qwen). They find that synthetic CoTs generated by stronger models often yield better downstream performance than human-written ones. They attribute this to distributional alignment and  partial correctness . They further validate these hypotheses via paraphrasing human traces to align their style with model-generated ones.

**Strengths:**

a.This paper emonstrates that distributional closeness can outweigh final-answer correctness is both surprising and practically significant for data curation strategies.

b.Multiple models with different parameters are evaluated.

c.Results consistently show that synthetic CoTs outperforms human-written CoTs.

d.The authors validate their hypothesis by paraphrasing the human-written CoTs.

**Weaknesses:**

a. The use of abbreviations like L-8B and W (for wrong CoTs) hinders immediate comprehension. The authors should adopt more explicit and standard notation, such as $\mathbf{Llama-8B}$ and "Incorrect CoTs" or "Wrong CoTs", to improve readability.

b. The training dataset is relatively small (under 1,000 examples) and restricted to the mathematical domain. This raises questions about the generalizability of the conclusions to other reasoning domains (e.g., commonsense, logical deduction) and larger-scale SFT.

c. The comparison between human-written and synthetic CoTs may be confounded by stylistic differences (e.g., length, verbosity, level of detail) rather than solely correctness. Human-written traces are noted as being significantly shorter, which could be an independent variable.

d. It remains unclear if this strong conclusion holds true across other benchmarks or against human-written CoTs that are confirmed to be of a much higher baseline quality.

**Questions:**

How does the quality of wrong CoTs influence performance after SFT? Potential factors of quality are response length, proportion of correct reasoning steps and clarity.

---

> ### Author Response · Authors · 2025-11-21
> **Author Response (1/1)**
>
> **[1] The use of abbreviations like L-8B and W (for wrong CoTs) hinders immediate comprehension. The authors should adopt more explicit and standard notation, such as  and "Incorrect CoTs" or "Wrong CoTs", to improve readability.**
>
> We will definitely make these changes in the final version of the paper to improve readability.
>
> **[2] The training dataset is relatively small (under 1,000 examples) and restricted to the mathematical domain. This raises questions about the generalizability of the conclusions to other reasoning domains (e.g., commonsense, logical deduction) and larger-scale SFT.**
>
> We agree that generalization to other domains and scales is important. However, our SFT datasets are substantially larger than 1,000 examples (see Table 8): approximately 6,000 problem–solution pairs for MATH and 7,000 for Countdown. For GSM8K, to keep G and W matched in size, we subsampled H and G to about 600 examples; W is smaller because GSM8K is easier and models produce a higher fraction of correct CoTs, reducing the pool of incorrect traces. Our study focuses on mathematical reasoning; we avoid claims beyond this domain. We will release code and data to support replication at larger scales and plan to extend to commonsense and logical-deduction tasks in future work.
>
> **[3] The comparison between human-written and synthetic CoTs may be confounded by stylistic differences (e.g., length, verbosity, level of detail) rather than solely correctness. Human-written traces are noted as being significantly shorter, which could be an independent variable.**
>
> We thank the reviewer for the interesting comments. Qualitatively, we sampled 10 CoTs from each of the H, G, and W datasets corresponding to G-2B setup, and we didn’t find any major stylistical differences - the structure and style remain consistent. Additionally, we present the length statistic in terms of the number of tokens for each of the datasets:
> - H: Avg tokens:  `227.0`
> - G: Avg tokens:  `216.3`
> - W: Avg tokens:  `284.1`
>
> We do observe a slight but not significant difference in the length of the CoTs as compared to the H dataset. However, a detailed study of these characteristics is an interesting future direction, and we thank the reviewer for highlighting the same.
>
> **[4] It remains unclear if this strong conclusion holds true across other benchmarks or against human-written CoTs that are confirmed to be of a much higher baseline quality.**
>
> We would also want to clarify that the Human CoTs here are of the best quality, totally correct but distributionally further from the base models. The results are not due to the fact that the Human CoTs have any contaminations or flaws. We are happy to run more experiments on other benchmarks that the reviewer suggests are of higher quality given our time and compute constraints. We would like to highlight that the benchmark should contain human annotated CoTs that are not written using language models to be valid for our study.
>
> **[5] How does the quality of wrong CoTs influence performance after SFT? Potential factors of quality are response length, proportion of correct reasoning steps and clarity.**
>
> We have addressed the point in the weaknesses above. We would like to highlight that certain characteristics, such as the proportion of correct reasoning steps, are hard to quantify at scale. However, we do a qualitative analysis of a subset for the same in Section 5.5.
>
> -----------------------------------------------------
> We would like to gently remind the reviewer to let us know if there are any further questions or feedback.
>
> We also respectfully request that the reviewer reconsider the score in light of the clarifications we have provided. We remain happy to engage in further discussion to address any additional concerns or questions.

---

### Official Review · Reviewer_QXaD · 2025-10-31

**Soundness:** 4
**Presentation:** 4
**Contribution:** 4
**Rating:** 8
**Confidence:** 4

**Summary:**

This paper challenges the conventional assumption that correctness is the primary determinant of data quality in reasoning tasks for LLMs, proposing instead that distributional similarity between synthetic data and the model's own output can be more critical. Through experiments on benchmarks like MATH, GSM8K, and Countdown, the authors demonstrate that training on model-generated chain-of-thought (CoT) traces with incorrect final answers (Category 3 data) can outperform human-annotated correct traces. The study also explores mechanisms like paraphrasing human CoTs to align with model distributions and gradually introducing flawed CoTs to quantify error tolerance, providing insights into how models learn from partially correct reasoning steps.

**Strengths:**

1）The paper directly questions the "correctness equals quality" dogma, suggesting that distributional proximity may outweigh final-answer accuracy. This idea opens new avenues for data curation, such as leveraging incorrect CoTs to avoid discarding useful signals, which is particularly valuable for resource-constrained scenarios in data-centric LLM research.

2）This paper designs rigorous experiments to validate their hypotheses. For instance, the paraphrasing experiment—where human CoTs are rewritten by an LLM to better match the model's distribution—shows improved performance, directly supporting the distribution similarity claim. Additionally, the error tolerance experiment progressively introduces fully flawed CoTs, revealing that models tolerate up to 25% errors without significant performance drop, highlighting their ability to extract signals from localized mistakes.

**Weaknesses:**

1）Experiments are confined to models ranging from 1.5B to 9B parameters, while modern LLMs often scale to hundreds of billions. Larger models may exhibit different sensitivities to noise or greater reliance on correctness, and the scaling behavior remains unexplored, limiting the generalizability of the findings.

2）Synthetic CoTs are generated by stronger models (e.g., Gemma-27B-IT), which could introduce biases. If the teacher model has systematic errors, the student model might amplify them, potentially compromising the reliability of the approach.

**Questions:**

1）On the Countdown task, G data (correct answers) clearly outperforms W data (incorrect answers), as shown in Figure 2, but the paper does not deeply investigate why distribution similarity fails to dominate in this case.

---

> ### Author Response · Authors · 2025-11-21
> **Author Response (1/1)**
>
> We thank the reviewer for their thoughtful comments. We address the comments below in more detail.
>
> **[1] Experiments are confined to models ranging from 1.5B to 9B parameters, while modern LLMs often scale to hundreds of billions. Larger models may exhibit different sensitivities to noise or greater reliance on correctness, and the scaling behavior remains unexplored, limiting the generalizability of the findings.**
>
> Due to time and compute constraints, we were unable to evaluate models beyond 9B; we note this in our limitations as well. Within the 1.5B–9B range, the observed trends are consistent, but we do not extrapolate to larger scales and view scaling behavior as important future work. We will release code and configurations to facilitate replication on larger models. We are also happy to run additional experiments suggested by the reviewers that fit within our compute capacity.
>
> **[2] Synthetic CoTs are generated by stronger models (e.g., Gemma-27B-IT), which could introduce biases. If the teacher model has systematic errors, the student model might amplify them, potentially compromising the reliability of the approach.**
>
> We agree that biases from teacher models can, in principle, transfer to student models. However, using LLM teachers is currently the only scalable and practical method for generating synthetic CoTs, so we ran our study in this setup. This remains an important problem for the community to mitigate. Empirically, our experiments test whether such bias transfer occurs in practice. First, across three datasets  and three student families, we consistently observe that SFT on synthetic CoTs, both correct and incorrect, improves downstream accuracy over human-written CoTs (refer Section 5.1). This indicates the student learns useful patterns; whether it also amplifies teacher errors is unclear from these experiments and warrants further study. Secondly, we directly probe sensitivity to systematic teacher errors by constructing datasets where up to 100% of the traces are deliberately flawed (fabricated formulas, inconsistent reasoning) and mixing them with correct traces at controlled ratios (refer Section 5.4). Models tolerate up to 25% flawed CoTs with performance comparable to or better than training on human data, and all synthetic datasets yield lower training loss than human-CoT datasets, indicating the student leverages distributional similarity rather than memorizing noise.
>
> **[3] On the Countdown task, G data (correct answers) clearly outperforms W data (incorrect answers), as shown in Figure 2, but the paper does not deeply investigate why distribution similarity fails to dominate in this case.**
>
> We would like to clarify this point in detail. Highly procedural and constrained tasks such as Countdown, unlike MATH and GSM8K, require a logically precise sequence of operations to reach the exact target. We hypothesize that learning signals from 'W' traces that lead to incorrect final expressions are weaker than “G” explaining the performance gap.
>
> -----------------------------------------------------
> We would like to gently remind the reviewer to let us know if there are any further questions or feedback.
>
> We also respectfully request that the reviewer reconsider the score in light of the clarifications we have provided. We remain happy to engage in further discussion to address any additional concerns or questions.

---

### Official Review · Reviewer_aeqo · 2025-11-01

**Soundness:** 3
**Presentation:** 2
**Contribution:** 2
**Rating:** 2
**Confidence:** 4

**Summary:**

The authors study the effect of model-generated vs human-provided chains-of-thought when fine-tuning LMs on mathematical reasoning tasks. They find that fine-tuning using synthetic CoTs leads to better model performance as compared to human-labeled CoTs. This still holds even if the synthetic CoTs are mistaken. The authors hypothesize that this is due to the fact that synthetic CoTs are closer in terms of distribution to the model's pretraining, which may make them more amenable for fine-tuning. They further show that using an LM to paraphrase human-provided CoTs can lead to further performance gains when fine-tuning models on the resulting dataset.

**Strengths:**

- The authors make an interesting finding that synthetic CoTs that lead to incorrect answers are more effective in SFT as compared to human-provided CoTs.
 - Experiments are conducted on a wide range of open language models, from different model families.

**Weaknesses:**

- The scope of the contributions is rather limited.
 - There are some methodological concerns (see below). For example, the statistical significance of the results is not reported.
 - There are some issues with the writing and the structure of the paper (see below). For example, there are numerous claims that are made without supporting references.

**Questions:**

There is a lot of background material that is not necessary to include for an ARR submission. For example, the background on autoregressive language modeling and SFT may be omitted or moved to the appendix. This would free up space for additional more novel contributions (such as the results referenced by Section 5.4).

The margins between the accuracies are quite small, and it is difficult to interpret the statistical significance of the results. Confidence intervals/bands would help to address this.

Why is Countdown considered out-of-distribution? Countdown is very similar to an earlier task called "24Game" which was mentioned in [https://arxiv.org/abs/2305.10601]. There, they used examples from the website [https://www.4nums.com/game/difficulties/] (which has existed since at least 2014 according to [web.archive.org]). It is very possible that the basic task in the Countdown game has been included in LLM pretraining data for many years.

Although the authors report starting training loss as a proxy for the difference in the distribution of the SFT data and the model's pretraining data, I am curious about the loss value *after* fine-tuning.

More detailed comments follow. While the paper is largely well-written, there are a handful grammatical and stylistic errors. I include some of them from the first three sections in the list below. This list is *not* comprehensive, so I encourage the authors to carefully read through the paper and correct all such errors.

Line 132: (minor) Missing space before citation.

Line 140: "This has motivated extensive study of how neural networks learn from noisy labels." This claim is missing support from references. Which studies?

Line 144: Use inline citations if the citation functions as a grammatical component in the sentence. In this case "Havrilla & Iyer (2024)" functions as a noun. In natbib, for example, \citet can be used for inline citations.

Line 153-155: The claims "It is observed that children do not merely mimic the linguistic data they are exposed to; instead, they often regularize it, producing a grammatical system that is more consistent and systematic than their input" and "Case studies, such as children of non-native speakers acquiring a more regular grammar than their parents" need support from references. Which studies?

Line 157: "evidences" -> "evidence"

Section 2.3 reads more like related work than as background.

Minor: "work" is traditionally an uncountable noun, and so it has no plural form.

Line 186: "Cots" -> "CoTs"

It is difficult to read the highly abbreviated names of the models.

---

> ### Author Response · Authors · 2025-11-21
> **Author Response (1/1)**
>
> We thank the reviewer for their comments. We address the comments below in more detail.
>
> **[1] The scope of the contributions is rather limited.**
>
> Please refer to the common response.
>
> **[2] There are some methodological concerns (see below). For example, the statistical significance of the results is not reported. The margins between the accuracies are quite small, and it is difficult to interpret the statistical significance of the results. Confidence intervals/bands would help to address this.**
>
> Please refer to our common response where we address your concern. To prove the statistical significance of our results, we replicate the experiments five times on Gemma 2B with the three main datasets H, G-27B-IT-W, and G-27B-IT-G and provide the mean and standard deviation, and confidence interval for more clarity. We are happy to address any other concerns.
>
> **[3] There are some issues with the writing and the structure of the paper (see below). For example, there are numerous claims that are made without supporting references.**
>
> We fixed all the issues. We already had the relevant citations but are re-adding them at more places for clarity.
>
> **[4] There is a lot of background material that is not necessary to include for an ARR submission. For example, the background on autoregressive language modeling and SFT may be omitted or moved to the appendix. This would free up space for additional more novel contributions (such as the results referenced by Section 5.4).**
>
> We would like to clarify that this is NOT an ARR submission. Although we see many prominent works that submit to ICLR having background material, we are happy to shift it to Appendix if the reviewer has serious concerns regarding the same.
>
> **[5] Why is Countdown considered out-of-distribution?...**
>
> We agree with the reviewer that calling Countdown “out-of-distribution” is a strong claim. We intended to convey that Countdown is a harder task than the other reasoning tasks we evaluate, given the ~0% zero-shot and very low few-shot accuracies of the base models (see Table 9). We have revised the paper accordingly and now describe Countdown as a “harder” task. We hypothesize that for highly procedural, constrained tasks like Countdown, the learning signal from “W” traces is weaker than from “G” traces, whereas this gap is smaller for more open-ended math problems. We discuss this in Section 5.1.
>
> **[6] Although the authors report starting training loss as a proxy for the difference in the distribution of the SFT data and the model's pretraining data, I am curious about the loss value after fine-tuning.**
>
> We report the loss curves in the Appendix. Please refer to Figures 6 (b), 8 (b), 10 (b) for the same. Clearly, the trends remain the same i.e. the human CoTs relatively remain farther from the model's distribution as compared to the synthetic ones across all the iterations.
>
> **[7] More detailed comments follow. While the paper is largely well-written, there are a handful grammatical and stylistic errors. I include some of them from the first three sections in the list below. This list is not comprehensive, so I encourage the authors to carefully read through the paper and correct all such errors.**
>
> We thank you for your comments. We have addressed the grammatical and stylistic errors.
>
> **[8] It is difficult to read the highly abbreviated names of the models.**
>
> We understand, and we’ll try to address this in the final version of the paper. We have already clearly defined our naming convention in the paper in section 4.
>
> -----------------------------------------------------
> We would like to gently remind the reviewer to let us know if there are any further questions or feedback.
>
> We also respectfully request that the reviewer reconsider the score in light of the clarifications we have provided. We remain happy to engage in further discussion to address any additional concerns or questions.

---

> ### Comment · Reviewer_aeqo · 2025-11-26
>
> I appreciate the authors for their response and for their suggested revisions, including the addition of confidence intervals to the results.
>
> I acknowledge that I mistakenly referred to this paper as an ARR submission, but my comment is still valid for an ICLR submission. The background is still unnecessary and I highly recommend that the content is moved to the appendix in order to provide space for additional contributions. Overall, I still find the scope of the contributions in the main text to be somewhat limited. For example, to what extent to the findings generalize beyond MATH, GSM8K, and Countdown—for example, to other kinds of (non-mathematical) reasoning tasks? Moving the ablation study and/or other results from the appendix to the main text could also help.
>
> I still find the abbreviations of the model names to be difficult to read. I recommend the authors use the names such as "Gemma2-2B" (or even "Gemma-2B" would be good, so long as the authors define the model as an instance of Gemma2). Are any non-instruction-tuned models used in the experiments? If not, the "-IT" designation is unnecessary, and it would suffice to define "Gemma2-2B" or "Gemma-2B" as the instruction-tuned version of the model.
>
> The presentation of the paper needs further improvement. For example, readers should be able to understand the figures at a high-level without needing to refer to definitions or details in the main text. As one (non-comprehensive) suggestion, I would recommend the legend labels in the figures to be changed into a non-abbreviated form (e.g., "Human-written CoT" rather than "H").
>
> If "3" were an option, I would raise my overall score to a 3, based on the revisions so far.

---

> > ### Author Response · Authors · 2025-12-02
> > **Author Response**
> >
> > **Re: Scope of Contributions**
> >
> > We have added new experiments on code generation which strongly support our findings, as detailed in the common response. We confirm that these new results (currently in Appendix A.9) will be moved to the main paper in the final version.
> >
> > **Re: Presentation and Clarity**
> >
> > We thank the reviewer for the suggestions to improve the paper's presentation. We will fully incorporate them into the final version. We are only deferring these specific edits for now to avoid re-numbering sections and figures, which could cause confusion during the discussion period. We commit to making the following changes:
> > - Move the background section to the appendix to make space for our new results.
> > - Revise all model abbreviations to the clearer format suggested.
> > - Use full, descriptive labels in all figure legends and tables.
> >
> > Based on our comprehensive rebuttal, we are confident that we have addressed all the reviewers' questions. We hope the reviewer agrees that these improvements further strengthens the paper and thus, will consider re-evaluating their score accordingly.

---

### Author Response · Authors · 2025-11-21
**Common Response by Authors**

We thank the reviewers for their helpful comments. We are happy that the reviewers find our work “interesting”, “valuable”, “surprising”, with our experiments being “rigorous”.

We would like to clarify some general points and reiterate the contributions of our paper for all the reviewers, ACs, and SACs.

We believe our work offers a useful insight to the community by demonstrating that distributional alignment is critical in reasoning tasks. Contrary to standard pipelines, which assume correctness is the primary determinant of data quality, we empirically establish that supervised finetuning (SFT)  on synthetic CoT traces, even those leading to incorrect final answers, can improve downstream performance, outperforming human-annotated "gold" data. We isolate this effect by showing that paraphrasing human data to become closer to the model’s output distribution induces better gains, confirming that the distributional gap in human text often hinders learning. The other factor is the existence of useful reasoning sub-steps in all the CoTs, even the ones with final incorrect answers, which provide the models enough information for reasoning improvement. Our findings reveal that the widely discarded "negative" data in current pipelines actually contains high-value, valid reasoning primitives, and we quantify model robustness to show tolerance for up to 25% completely flawed data before performance degrades. Our work is novel in prioritizing the distribution of the CoTs as measured using the model being finetuned over final-answer correctness, offering a scalable paradigm for improving reasoning that reduces the reliance on expensive human annotation.

### Significance Testing

We report mean $\pm$ standard deviation and 95\% confidence intervals on the mean for robustness metrics over 5 random seeds. The experiments were run using Gemma 2B (G2B) over the datasets: H, G27B-IT-G, and G27B-IT-W.

| Run | **Mean $\pm$ Std Dev** | **95% CI on Mean** |
| :--- | :--- | :--- |
| G2B on H | $0.162 \pm 0.005$ | $0.162 \pm 0.006$ |
| G2B on G27B-IT-W | $0.183 \pm 0.008$ | $0.183 \pm 0.009$ |
| G2B on G27B-IT-G | $0.223 \pm 0.005$ | $0.223 \pm 0.006$ |


### Ablations Across Different Hyperparameters

We perform a detailed ablation of the main hyperparameters - batch size (BS) and learning rate (lr) - over the Gemma-2B model on the MATH related datasets to show that the trends of results that synthetic G and W CoTs outperform H CoTs remain consistent across all runs. We provide maximum accuracy on MATH500 test set for G-2B for all these runs.


| Model | **H** | **G-27B-IT-W** | **G-27B-IT-G** |
| :--- | :--- | :--- | :--- |
| **BS 256 lr 2e-5** | 0.17 | +0.02 | +0.04 |
| **BS 256 lr 1e-6** | 0.14 | +0.03 | +0.05 |
| **BS 64 lr 2e-5** | 0.17 | +0.02 | +0.06 |
| **BS 64 lr 1e-6** | 0.15 | +0.02 | +0.04 |
| **BS 16 lr 2e-5** | 0.15 | +0.02 | +0.05 |
| **BS 16 lr 1e-6** | 0.17 | +0.02 | +0.02 |

We have added additional paraphrasing experiments (Figure 13) and loss curves (Figure 14) to strengthen our findings.

---

### Author Response · Authors · 2025-12-02
**Common Response: Additional experiments on other reasoning tasks (code generation)**

We conducted additional experiments on a new task: code generation that strongly supports our findings and makes them generalizable. We curated 354 samples of H, G and W using the MBPP datasets {1}  and finetuned G-2B model. We evaluate the finetuned models using pass@1 with T=0. We observe 4% and 7% absolute accuracy increase wrt H baseline on the held out test data containing 200 problems using W and G dataset respectively (refer to the table below). Moreover, we performed the paraphrasing experiment using the H data and the performance improved by 6% absolute increase. The loss curves shown in Figure 15 (b) and Figure 16 (b) show that H dataset is distributionally farther away from the model’s distribution as compared to the G and W dataset, strongly supporting our finding: synthetic CoTs leading to correct or incorrect final answers can outperform entirely correct human written CoTs if the synthetic CoTs are closer to the model's distribution. We also provide the prompt for this code solution paraphrasing experiment below. Details of the experiments are presented in Appendix A.9.


| Model  | H    | G-27B-IT-W | G-27B-IT-G | H-Para-G-27B-IT |
|--------|------|------------|------------|------------------|
| `G-2B` | 0.22 | +0.04      | **+0.07**  | +0.06           |


{1} Austin, Jacob, et al. "Program synthesis with large language models." arXiv preprint arXiv:2108.07732 (2021).

---

### Author Response · Authors · 2025-12-02
**Final Common Response**

We thank the reviewers for their useful comments that strengthened our paper. We would like to highlight the following key points:

**Novelty**: We present an interesting and new finding that synthetic CoTs leading to incorrect final answers can outperform entirely correct human written CoTs if the synthetic CoTs are closer to the model's distribution. We conduct several comprehensive experiments on 4 reasoning tasks including math, algorithm reasoning and code generation, and present quantitative and qualitative insights to support our findings.

We highlight the importance of “distribution” of the datasets to improve the model’s performance with SFT. While prior work (as correctly pointed out by Reviewer BDoS) has shown the importance of reasoning structure, our work identifies a distinct and more general principle: dataset distribution.

**Additional Experiments**: During the discussion period, we tried to address every concern of the reviewers and conducted experiments as suggested:

- **New Domain Generalization**: Added a full set of experiments on code generation (MBPP), which strongly supports our main findings.
- **Statistical Significance**: Added confidence intervals for all main results to confirm the statistical significance of our findings.
- **Hyperparameter Ablations**: Performed new ablations to test the sensitivity of our results to different hyperparameters.
- **Additional Loss Analysis & Paraphrasing**: Added more loss plots and expanded paraphrasing experiments (for L-8B) to further support our distribution hypothesis.
- **Quantitative Comparison of Different CoTs**: In addition to the qualitative analysis of H, W and G CoTs, we also compute perplexity and length analysis of the CoTs indicating their similar structure but different distribution.

Based on our comprehensive rebuttal, we are confident that we have addressed all the reviewers' questions. We hope the reviewers agree that these improvements strengthen the paper and will consider re-evaluating the original scores accordingly.

---

### Note · Program_Chairs · 2026-01-17
**Submission Desk Rejected by Program Chairs**

The following references in this submission do not refer to real documents and/or have major errors in bibliographic information:

 Gabriel Bercovich, Swaroop Mishra, Gaurav Singh Tomar, Rajarshi Das, Chunyuan Li, Naveen Jafer, Baolin Peng, Yuxiong He, Mihir Kale, and Jianmo Ni. Llama-nemo: An efficient reasoning model family through unified reward function and progressive-rl, 2025.